# MeCP2 regulates gene expression through recognition of H3K27me3

Wooje Lee [1], Jeeho Kim [1], Jung-Mi Yun [2], Takbum Ohn [1✉] & Qizhi Gong [3✉]

MeCP2 plays a multifaceted role in gene expression regulation and chromatin organization. Interaction between MeCP2 and methylated DNA in the regulation of gene expression is well established. However, the widespread distribution of MeCP2 suggests it has additional interactions with chromatin. Here we demonstrate, by both biochemical and genomic analyses, that MeCP2 directly interacts with nucleosomes and its genomic distribution correlates with that of H3K27me3. In particular, the methyl-CpG-binding domain of MeCP2 shows preferential interactions with H3K27me3. We further observe that the impact of MeCP2 on transcriptional changes correlates with histone post-translational modification patterns. Our findings indicate that MeCP2 interacts with genomic loci via binding to DNA as well as histones, and that interaction between MeCP2 and histone proteins plays a key role in gene expression regulation.

---

[1] Department of Cellular & Molecular Medicine, College of Medicine, Chosun University, Gwangju 61452, South Korea. [2] Department of Food and Nutrition, Chonnam National University, Gwangju 61186, South Korea. [3] Department of Cell Biology and Human Anatomy, University of California at Davis, School of Medicine, Davis, CA 95616, USA. ✉email: tohn@Chosun.ac.kr; qzgong@ucdavis.edu

Methyl CpG binding protein 2 (MeCP2) is highly expressed in neurons, where expression increases postnatally after the final stage of neurogenesis[1,2]. Mutations in the *MECP2* gene constitute the primary cause of the neurodevelopmental disorder Rett syndrome (RTT)[3–5]. It was hypothesized that MeCP2 functions as a transcriptional repressor that targets methylated DNA at CpG islands to repress transcription[6,7]. Using high throughput assays, several studies demonstrated that, in addition to its affinity to methylated CpG[2], MeCP2 is also localized to methylated cytosine in the non-CG context (mCH, where H = A, C, or T) and to many non-methylated loci[8–12].

We recently found that MeCP2 localization can be predicted by genomic DNA sequence features[13], particularly regions of high GC content where intrinsic nucleosome occupancy is observed[14,15]. Several studies, including ours, demonstrated that MeCP2 co-localized with nucleosomes or the nucleosome associated protein, histone H1[13,16,17]. In addition, in native co-immunoprecipitation (co-IP) assays, MeCP2 binds to H3K9/H3K27 methylated nucleosomes in brain tissue[18]. Moreover, in pancreatic adenocarcinoma cell lines, MeCP2 is associated with H3K9 methylation in the *IL6* gene region[19]. These studies suggest that MeCP2 affinity to chromatin can be achieved through binding to DNA as well as to histones.

In this study, we demonstrate that MeCP2 can be recruited to genomic loci via binding to H3K27me3 and co-enrichment of MeCP2 and H3K27me3 at transcription start site (TSS) cooperatively regulates gene expression. Our findings indicate that MeCP2, in addition to direct DNA binding, can interact with chromatin via histone protein and that MeCP2 interaction with histone post-translational modifications (PTMs) plays significant roles in transcription regulation.

## Results

**MeCP2 is associated with nucleosomes in vivo.** MeCP2 interacts with several nuclear proteins including DNMT1, CoREST, Suv39H1, and c-SKI[20–22]. To further identify and characterize interacting proteins, MeCP2 was immunoprecipitated (IP) from an olfactory epithelium (OE) nuclear extract of wild-type (WT) mice with MeCP2 specific antibody (Supplementary Fig. 1A). Three distinct bands were identified among co-IPed proteins. One is at ~28 kDa (star in Fig. 1a), and two other bands are found between 10 and 15 kDa (arrowheads in Fig. 1a). To identify the 28 kDa band, the band was excised from the gel and analyzed using MALDI-TOF mass spectrometry (Supplementary Fig. 1C). The sequence of the tryptic peptides, determined by MALDI-TOF, aligns significantly with histone H1 (H1) isoforms (Supplementary Fig. 1D). To confirm the MeCP2 and H1 interaction, we performed MeCP2 IP using Benzonase treated nuclear extracts (Supplementary Fig. 1B) and validated the presence of H1 (Fig. 1b). We further performed reverse co-IP using a pan H1 antibody. MeCP2 was identified in the H1 co-IPed pool of proteins, determined by Western blotting (Fig. 1b). The other two proteins co-IPed with MeCP2 correspond to histone subunits, histone H3 (H3) and histone H4 (H4), in terms of their molecular weights (arrowheads in Fig. 1a). To test whether MeCP2 is also associated with other histone subunits, the co-IPed proteins from Benzonase treated nuclear extracts were examined for the presence of H3 and H4 by western blotting. Both H3 and H4 were found to be co-precipitated with MeCP2 (Fig. 1c).

With recent findings that MeCP2 and H1 often coexist in the same genomic regions[17] and MeCP2 preferentially locates at GC-rich regions[13], which are populated with nucleosomes[14,15], we hypothesized that MeCP2 is located in genomic loci through binding to histone proteins. To determine the genome-wide distribution of MeCP2, nucleosomes and H1, we performed H1 chromatin immunoprecipitation sequencing (ChIP-seq) using a pan H1 antibody (Supplementary Fig. 4A) and compared the results to those of a high-resolution MeCP2 ChIP-seq and MNase-seq data sets[13] (Fig. 1d–f). We compared local enrichment of H1 and nucleosome locations, determined by MNase-seq, to MeCP2-enriched loci on chromosome 19 (chr 19) (Fig. 1d). Within the MeCP2 enriched loci, nucleosome localization and H1 binding demonstrated co-localization. Also, for nucleosome locations on chr 19 defined from MNase-seq, enrichment of MeCP2 and H1 showed high degrees of overlap (Fig. 1e). In a genome-wide correlation analysis between MeCP2 and H1, we observed that the MeCP2 ChIP-seq coverage correlated with that of H1 ChIP-seq genome-wide ($r = 0.60$) (Fig. 1f). In addition, MNase-seq also moderately correlates with MeCP2 ChIP-seq ($r = 0.58$) and with H1 ChIP-seq ($r = 0.54$).

To further confirm that H1 and MeCP2 are localized at the same loci, we performed ChIP-qPCR experiment for selected regions in the 5′UTR of the *Bdnf* gene, using both H1 and MeCP2 ChIPed DNA. Co-enrichment of H1 and MeCP2 binding was observed for all selected regions tested (Supplementary Fig. 3A,B). To further confirm the observed co-presence was due to co-occupancy of the same binding site by both MeCP2 and H1, we performed sequential ChIP analysis. MeCP2 ChIPed DNA was subsequently ChIPed with a pan H1 antibody. The resulting DNA was evaluated with qPCR. The re-ChIP analysis confirmed the occupancy of both MeCP2 and H1 in the 5′UTR of the *Bdnf* gene (Supplementary Fig. 3A, C). These results altogether indicate that MeCP2 physically associated with the nucleosomal complex.

**MeCP2 is highly co-localized with H3K27me3.** Despite the significant correlation between MeCP2 and nucleosomes throughout the genome (for example, chr 18: 80,961,548–80,969,056, Fig. 2a center), detailed analysis shows that MeCP2 and nucleosomes do not always co-localize. For example, within chr 2: 109,320,845–109,328,734 (Fig. 2a left), the nucleosome structure is distinct and MeCP2 binding is low, while within chr 17: 78,596,919–78,604,752 (Fig. 2a right), MeCP2 binding is robust but nucleosome structure is less stable. To better determine the frequency of MeCP2 and nucleosome colocalization, we utilized the PING software[23,24]. The PING algorithm is designed to detect nucleosome locations with MNase-seq or sonicated-ChIP-seq data. We detected ~7672k nucleosome locations and 4257k MeCP2 binding peaks (Fig. 2b). Among those, MeCP2 was absent from 5291k nucleosome locations (68.9% of total nucleosomes), and 2382k nucleosome locations (31.0% of total nucleosomes) were MeCP2-occupied. We also found that 1,876k MeCP2 peaks (44% of total MeCP2 peaks) were observed in nucleosome-free regions. To account for these deviations, the level of DNA methylation was assessed in the following three types of regions: MeCP2-absent mononucleosome loci (group1), MeCP2-enriched mononucleosome loci (group2), and nucleosome-free MeCP2-binding loci (group3). We randomly selected 10,000 loci from each category and compared the methylation status of CpG, CAG, and CAH with whole-genome bisulfite sequencing (WGBS)[13] (Fig. 2c). For the cytosine methylation in CpG, the MeCP2-bound loci were significantly more frequently methylated (median methylation = 50.0% in group1, 76.2% in group2, and 73.9% in group3). In comparison with cytosine methylation in tri-nucleotides (CAG and CAH), the loci from each group were mostly hypo-methylated (median methylation = 0.00% and 0.00% in group1, 0.011% and 0.004% in group2, and 0.009% and 0.006% in group3 for CAG and CAH, respectively). However, the methylation status of the MeCP2 bound loci was still significantly higher than that of MeCP2

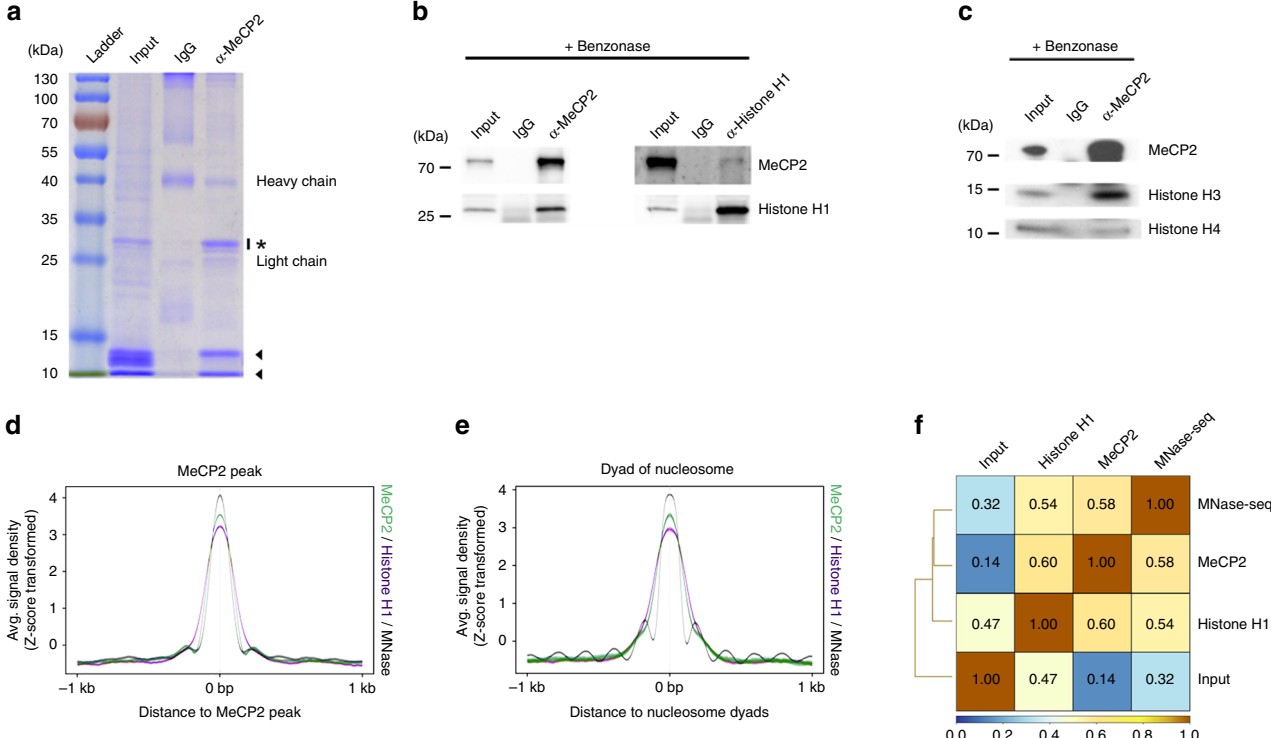

**Fig. 1 MeCP2 is physically associated with nucleosome complexes. a** MeCP2 antibody immunoprecipitated proteins from mouse OE nuclear extract in SDS gel visualized with Coomassie blue. Nuclear extract is loaded as input; rabbit IgG as a negative control. Star is around 25–30 kDa. **b** Benzonase treated nuclear extracts were immunoprecipitated either with anti-MeCP2 antibody or anti-histone H1 antibody. Samples of input, IgG and immunoprecipitates were analyzed with western blotting for histone H1 or MeCP2. **c** Benzonase treated nuclear extracts were immunoprecipitated with a MeCP2 antibody specific to its C-terminus. Input, IgG control and immunoprecipitates were analyzed with Western blotting for the presence of histone H3 and H4. **d** Z-transformed aggregate plot of the average tag density of MeCP2 binding sites (hotspot, green) at chr19 occupied by: Nucleosome occupancies (gray) and histone H1 (purple). **e** Z-transformed aggregate plot of the average tag density of MNase-seq (hotspot, gray) at chr19 occupied by: MeCP2 (green) and histone H1 (purple). **f** Genome-wide inter-correlation among Input, histone H1 ChIP-seq, MeCP2 ChIP-seq, and MNase-seq (binning size = 150 bp). Number in the square indicates the Pearson correlation coefficient. Source data are provided as a Source data file (**b** and **c**). The shaded areas are up to ±S.D. from the average profile (**d** and **e**).

non-bound loci. Consistent with previous studies, MeCP2 exhibits affinity to methylated DNA[2,8,9].

To examine the relationship between post-translationally modified histones and MeCP2 enrichment, we performed ChIP-seq for H3K27me3 and H3K9ac using the same genomic DNA used for MeCP2 ChIP-seq[13] (Supplementary Fig. 4B and C). Among chr19 loci, when mononucleosomes are clearly present, MeCP2 levels (group1, $n = 101,549$; group2, $n = 67,051$) correlated with the levels of H3K27me3. When nucleosomes were unstable (group3, $n = 33,644$), MeCP2 enrichment also showed a strong correlation with H3K27me3 while H3K9ac levels are high (Fig. 2d). These results indicate that MeCP2 binding is correlated with H3K27me3.

We also analyzed statistical differences in the histone PTM between the groups (Fig. 2e, f). In MeCP2-enriched loci (median H3K27me3 [log10] of group2 = −0.51, and group3 = −0.51), regardless of nucleosome formation, H3K27me3 was significantly more abundant than that in group1 (median H3K27me3 [log10] = −4.00) (Fig. 2e). When compared to group1 (median H3K9ac [log10] = −4.00), H3K9ac signals in group2 regions (median H3K9ac [log10] = −1.41) are significantly higher (Fig. 2f). Furthermore, H3K9ac signals in group3 (median H3K9ac [log10] = −1.41) are significantly higher than that of the group2. At the genomic level, the correlation of MeCP2 enrichment with H3K27me3 is higher than that with H3K9ac (r for H3K27me3 = 0.74, H3K9ac = 0.27) (Fig. 2g). Overall, the MeCP2 enrichment to chromatin is highly correlated with H3K27me3.

**MeCP2 binds to nucleosomes containing H3K27me3 via MBD.** To test that MeCP2 preferentially binds to H3K27me3, we performed an in vitro pull-down assay to examine whether purified MeCP2 exhibits higher affinity to H3K27me3 containing nucleosomes compared to unmodified nucleosomes. Purified GST-tagged MeCP2 proteins were incubated with either biotin-tagged unmodified mononucleosomes or H3K27me3 modified mononucleosomes (Fig. 3a). We observed that MeCP2 was more efficiently pulled-down by H3K27me3 modified mononucleosomes, indicating stronger MeCP2 affinity to H3K27me3. This is consistent with our observation that MeCP2 and H3K27me3 can be co-IPed using neuronal tissue (Supplementary Fig. 5C).

To further investigate which domain of MeCP2 plays a central role for this interaction, we performed structure-binding experiments by expressing Flag-tagged full length or truncated MeCP2 in 293T cells (Fig. 3b). Flag-tagged MeCP2 and mutants were immunoprecipitated and histone H3 was evaluated by Western blotting (Fig. 3c). Full-length MeCP2 and mutants containing methyl-CpG-binding domain (MBD) efficiently immunoprecipitated histone H3, while the MBD lacking fragments fail to pull down histone H3. We further tested whether MBD of MeCP2 preferentially interact with modified histone H3, H3K27me3. Interactions between bait proteins (purified GST, GST-tagged MeCP2 or MBD) and prey proteins (recombinant mononucleosomes, either containing unmodified H3 or H3K27me3) were evaluated by pulldown analysis with glutathione affinity column (Fig. 3d and Supplementary Fig. 5A). Both full length MeCP2 and

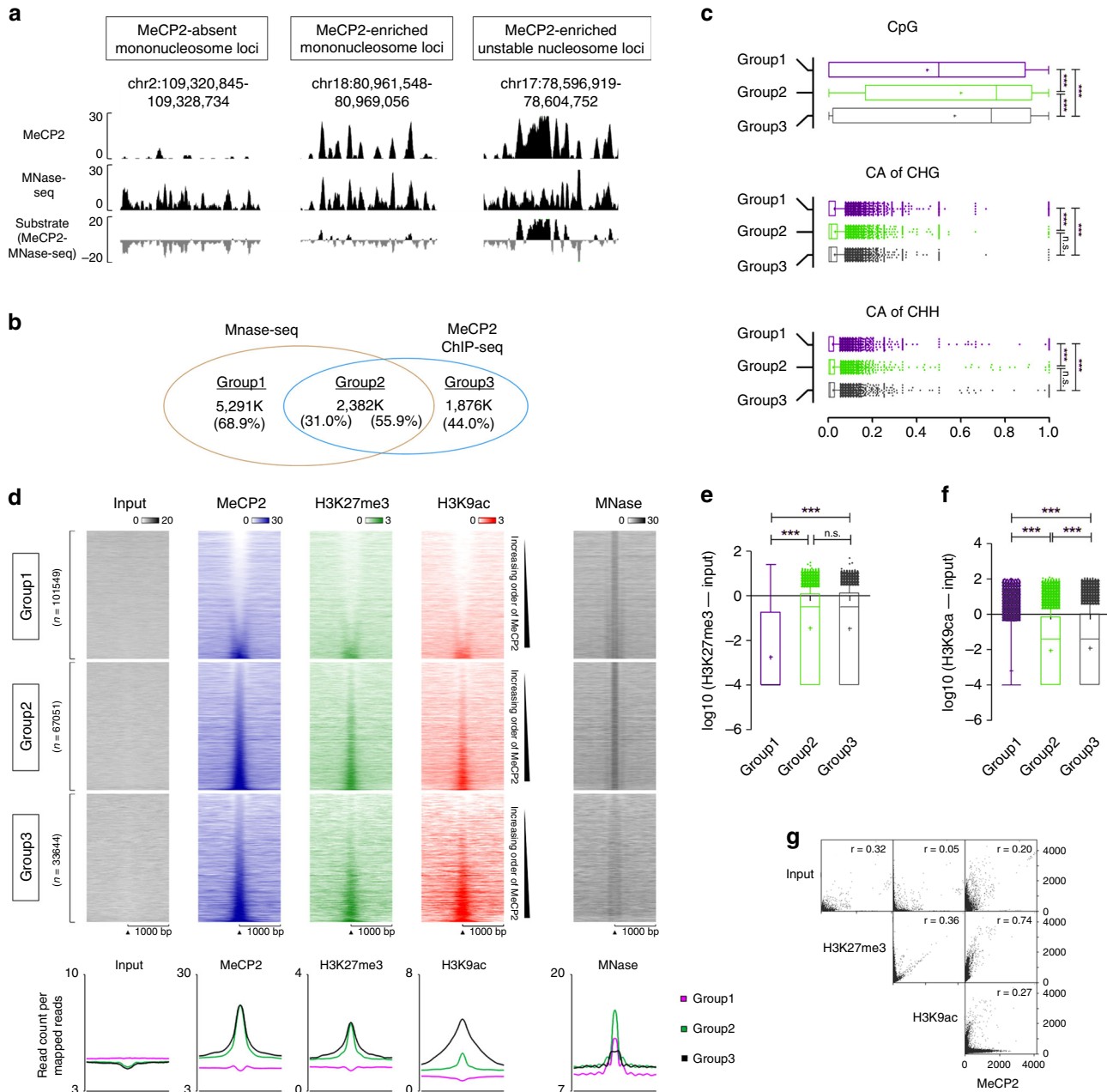

**Fig. 2 Epigenetic characteristics of MeCP2 binding or non-binding mononucleosomes. a** Examples of MeCP2 ChIP-seq and MNase-seq tracks for MeCP2-absent mononucleosome occupied loci (left), MeCP2-enriched mononucleosome occupied loci (center), and MeCP2 binding in unstable nucleosome loci (right). **b** Venn diagram showing the number of peaks in MeCP2-absent mononucleosome loci (group1), MeCP2-enriched mononucleosome loci (group2) and nucleosome-free MeCP2-binding loci (group3). **c** Comparison of methylation levels of CpG, CAG, and CAH in each of 10,000 loci randomly selected from the three groups defined in **b**. $N = 2$ (WT) from biologically independent mice. For CpG, $p = 1.11 \times 10^{-115}$ (group1 vs group2); $p = 2.4 \times 10^{-85}$ (group1 vs group3); $p = 1.16 \times 10^{-4}$ (group2 vs group3). For CAG, $p = 1.06 \times 10^{-75}$ (group1 vs group2); $p = 1.55 \times 10^{-63}$ (group1 vs group3); $p = 0.17$ (group2 vs group3). For CAH, $p = 1.69 \times 10^{-22}$ (group1 vs group2); $p = 9.85 \times 10^{-27}$ (group1 vs group3); $p = 1.08$ (group2 vs group3). **d** Heatmap of Input, MeCP2, H3K27me3, H3K9ac ChIP-seq peaks and MNase-seq with flanking regions, of Chr19 by the types of regions defined in **b**. Aggregate plot of average Input, MeCP2, H3K27me3, H3K9ac and MNase-seq reads below. **e, f** Comparison of H3K27me3 (**e**) or H3K9ac (**f**) levels among the three groups of loci at chr19 (group1, $n = 101,549$; group2, $n = 67,051$; group3, $n = 33,644$). $N = 2$ (WT) from biologically independent mice. For H3K27me3, $p < 0.0001$ (group1 vs group2); $p < 0.0001$ (group1 vs group3); $p = 0.02$ (group2 vs group3). For H3K9ac, $p < 0.0001$ (group1 vs group2); $p < 0.0001$ (group1 vs group3); $p = 2.34 \times 10^{-33}$ (group2 vs group3). $p = 0.00$ were reported as $p < 0.0001$. **g** Scatter plots showing inter-correlation between H3K27me3 and H3K9ac ChIP-seq and MeCP2 enrichment (binning size = 1000 bp). Two-tailed Mann–Whitney testing with Bonferroni corrections, ***$p < 0.0001$ (**c, e**, and **f**). Box-and-whisker plots show median, 10 to 90 percentile, and min and max values (**c, e**, and **f**). Mean is marked by "+."

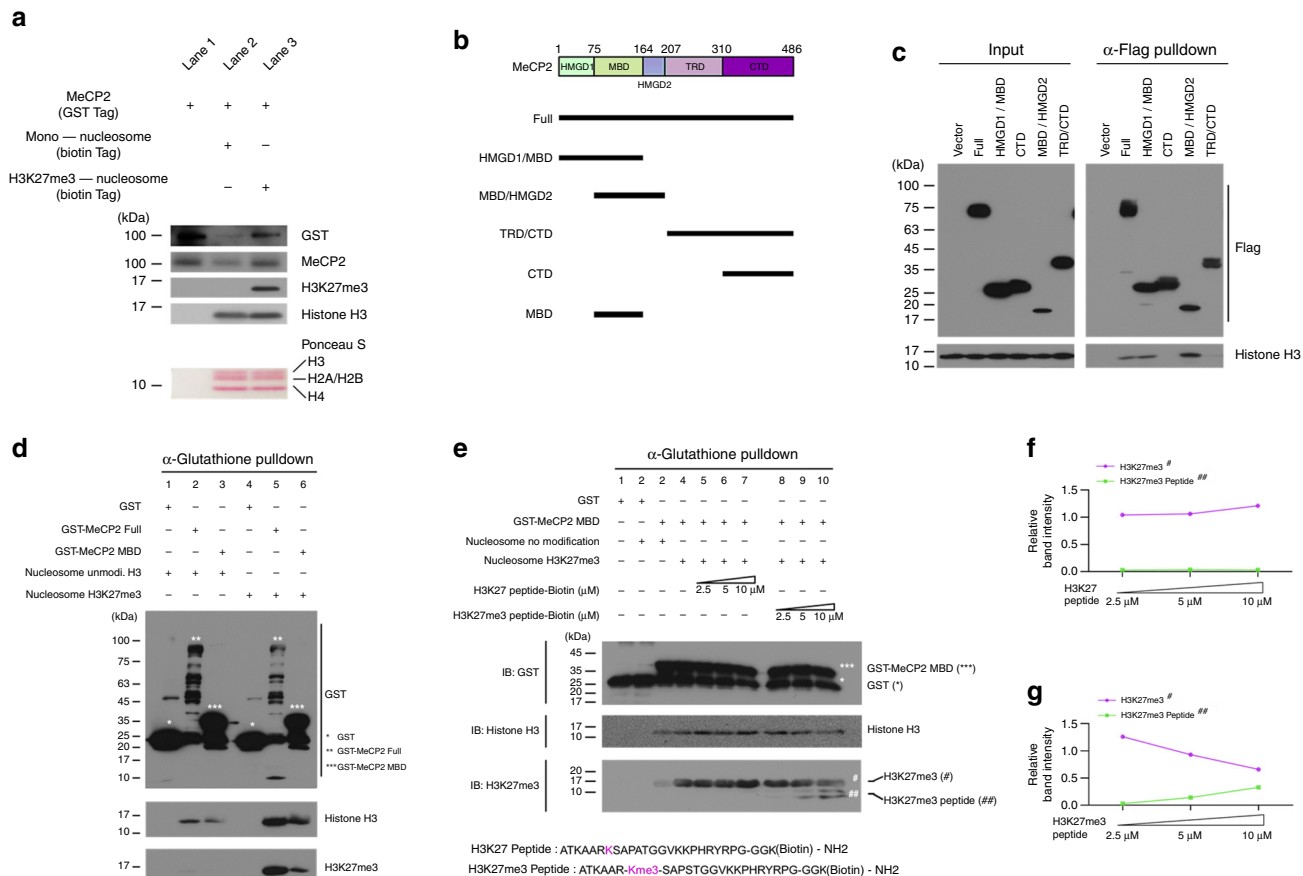

**Fig. 3 MeCP2 binding to nucleosome is enhanced by the presence of H3K27me3. a** Interactions between MeCP2 and H3K27me3 are determined by in vitro pull-down assay. Pull-down MeCP2 is shown by both MeCP2 and GST bands. Comparable levels of mononucleosomes between land 2 and 3 are shown by Ponceau S staining and H3. The presence of H3K27me3 is validated in Lane 3. **b** Schematic representation of MeCP2 domain constructs. **c** Co-immunoprecipitation of histone H3 with Flag-tagged MeCP2 full length or truncated fragments. From 293T cell transfected with Flag-tagged constructs in **b**, total protein extract was immunoprecipitated with Flag antibody and the presence of histone H3 is shown by immunoblotting. **d** In vitro pull-down assays of bait (purified GST, GST-tagged full length MeCP2 or MBD of MeCP2) to prey protein (recombinant mononucleosomes, either containing unmodified H3 or H3K27me3). Pull-downed H3 or H3K27me3 were visualized by SDS-PAGE and immunoblotting with anti-Histone H3 or H3K27me3. The presence of GST-fusion protein in each construct was indicated by the asterisks. **e** Competition assays with unmodified H3K27 or H3K27me3 peptides to MBD/nucleosome interactions. GST-tagged MBD binding to nucleosomes (top blot) containing H3K27me3 (lane 4) were challenged with H3K27 (lane 5–7) and H3K27me3 (lane 8-10) peptide. Bindings of GST-tagged MBD to histone H3 (middle blot) and H3K27me3 (bottom blot, sharp) were evaluated by GST pull-down and immunoblotting. H3K27me3 peptides in the MBD/nucleosome complexes were also revealed (bottom blot, double sharp). **f**, **g** Quantification of normalized H3K27me3 (purple, sharp) and H3K27me3 peptide (green, double sharp) levels by densitometry analysis in **e**. The signal strength in each band is normalized to H3K27me3 signal strength lane 4 in **e**. Source data are provided as a Source data file (**a**, **c**, **d** and **e**).

MBD showed higher affinity to mononucleosomes containing H3K27me3 (Fig. 3d, lane 5 and 6) compared to those with unmodified H3 (Fig. 3d, lane 2 and 3).

The preference of MBD binding to H3K27me3 rather than unmodified H3 was further investigated by the competition assay. Purified MBD fragments were incubated with nucleosomes containing H3K27me3 (Fig. 3e). H3K27 or H3K27me3 peptides were added and competitive bindings were evaluated. As concentration of H3K27me3 peptide increased, the binding of GST-tagged MBD to nucleosomes containing H3K27me3 decreased. In the meantime, MBD binding to H3K27me3 peptide correspondingly increased (Fig. 3e, g, and Supplementary Fig. 5B). To further substantiate the specificity in the competition experiment, a peptide with the same sequence of H3K27, but contained no modification, was tested in parallel (Fig. 3e). We observed no obvious competitive binding between H3K27 peptide to MBD-nucleosome complex even at the highest concentration, 10 μM, tested (Fig. 3e, f, and Supplementary Fig. 5B). Thus, comparing to unmodified H3, MBD of MeCP2 shows higher

affinity to H3K27me3. Taken together, these results indicate that MeCP2 directly binds to histone H3 via MBD and the binding affinity is preferential to H3K27me3.

Direct association between MeCP2 and H3K27me3 was further tested. If MeCP2 enrichment is directly bound to H3K27me3, it is predicted that an alteration in H3K27me3 levels would change the level of MeCP2 binding on the chromatin. To decrease the H3K27me3 levels, SH-SY5Y cells were treated with GSK343, an inhibitor of H3K27me3-specific histone methyltransferase (HMTase)[25]. After 72 hours of treatment, we observed that the level of H3K27me3 was dramatically decreased, while the level of Ezh2, an H3K27-specific histone methyltransferase, is unchanged (Supplementary Fig. 6A, B). In addition, the transcription and protein levels of MeCP2 were unchanged after GSK343 treatment (Supplementary Fig. 6A–C). Furthermore, we determined whether GSK343 treatment alters DNA methylation levels using bisulfite pyrosequencing of LINE-1 regions[26] (Supplementary Fig. 6D–F) and targeted bisulfite sequencing[27] (Supplementary Fig. 6G, H). Both results demonstrated that GSK343 treatment for

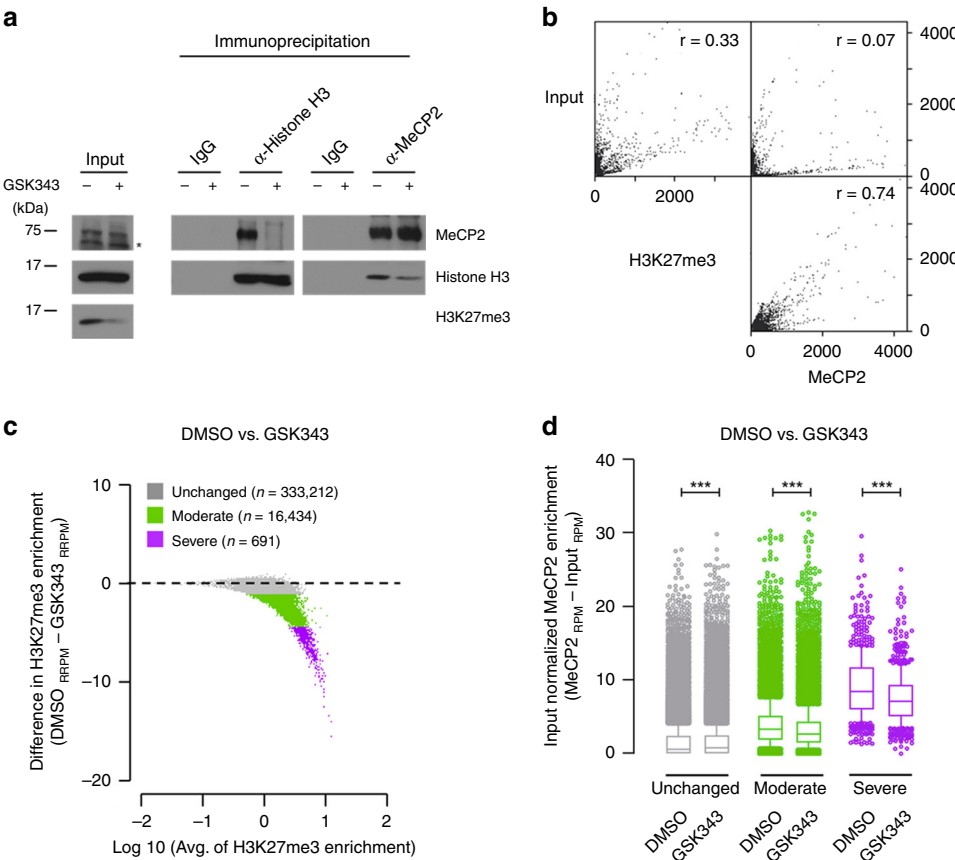

**Fig. 4 MeCP2 binding to chromatin is regulated by the levels of H3K27me3. a** SH-SY5Y cells treated with either DMSO or GSK343 were immunoprecipitated with either histone H3 or MeCP2 antibody. Samples of input were analyzed with Western blot for MeCP2, histone H3 and H3K27me3. IgG and immunoprecipitates were analyzed with Western blot for MeCP2 and histone H3. Star indicates a non-specific band. Source data are provided as a Source Data file. **b** Scatter plots show the correlation between H3K27me3 ChIP-seq signal and MeCP2 enrichment from SH-SY5Y cells (binning size = 1000 bp). $N = 2$ (WT) from biologically independent experiments. **c** Differences of exogenous reference normalized H3K27me3-enrichment between GSK343 treatment and control DMSO are plotted. $N = 2$ (per treatment) from biologically independent experiments. The x axis corresponds to the average of H3K27me3 signals in both GSK343 treated and DMSO control; the y axis corresponds to the difference of H3K27me3 signal between GSK343 treated and DMSO control SH-SY5Y cells. Based on the H3K27me3 difference, loci are categorized as unchanged (gray), moderate (green) or severe (purple). **d** MeCP2 enrichment of DMSO or GSK343-treated samples, separated by groups defined in **c**. $N = 2$ (per treatment, DMSO vs GSK343) from biologically independent experiments. $n = 333,212$, $p < 0.0001$ (Unchanged, gray); $n = 16,434$, $p < 0.0001$ (Moderate, green); $n = 691$, $p = 7.75 \times 10^{-51}$ (Severe, purple) (two-tailed Wilcoxon signed-rank test, *** $p < 0.0001$). $p = 0.00$ were reported as $p < 0.0001$. Box-and-whisker plots show median, 10–90 percentile, and min and max values.

72 h had little effect on DNA methylation. Given these findings, we performed immunoprecipitation assays to determine MeCP2 and H3K27me3 interaction. Input from the GSK343-treated cells show less H3K27me3 compared to the DMSO controls as expected (Fig. 4a). More importantly, the IPed histone H3 from GSK343-treated cells contained markedly less MeCP2 than that of DMSO-treated cells. Consistently, reverse co-IP by MeCP2 antibody in GSK343-treated cells showed much lower levels of H3 than that of DMSO-treated cells (Fig. 4a).

In parallel, whether MeCP2 binding to chromatin decreases with reduced H3K27me3 was examined by H3K27me3- and MeCP2 ChIP-seq of DMSO- and GSK343-treated cells (Fig. 4b to d). Consistent with in vivo analysis (Fig. 2g), MeCP2 enrichment in DMSO-treated SH-SY5Y cells shows strong correlation with H3K27me3 ($r = 0.74$). Differential peak calling analysis comparing DMSO- and GSK343-treated cells shows overall H3K27me3 peak reduction, while a fraction of MeCP2 peaks show variation in localization (Supplementary Fig. 7A, B). H3K27me3 signal was further normalized with an exogenous reference genome (ChIP-Rx) and determined by reference-adjusted reads per million (RRPM) (Supplementary Fig. 7C). We classified loci, according to

the degree of H3K27me3 level changes by GSK343 treatment after RRPM normalization: "unchanged", "moderate" and "severe" (Fig. 4c). The loci in the "moderate" and "severe" groups showed relatively high H3K27me3 enrichment compared the "unchanged" loci. We then compared MeCP2 enrichment after read per million (RPM) normalization by the groups (Fig. 4d). MeCP2 binding in the loci of the "unchanged" group showed a small but significant increase in the GSK343-treated samples compared to DMSO, despite the overall MeCP2 enrichment is low in those regions (median of MeCP2 differences = 0.00, Fig. 4d). Among the loci in the "moderate" group, MeCP2 binding is significantly reduced in GSK343-treated cells compared to that in the control DMSO (median of MeCP2 differences = −0.47, Fig. 4d). Moreover, in the "severe" group, in addition to significant reduction of MeCP2 binding, the degree of MeCP2 reduction (median of MeCP2 differences = −1.5, Fig. 4d) is at a greater degree compared to that of the "moderate" group. We further examined H3K27me3 and MeCP2 enrichment at three loci around the *HOXB* gene cluster (Supplementary Fig. 7F, G). A significant decrease in the MeCP2 binding upon H3K27me3 reduction was observed in these loci as well. Taken

together, these results suggest that MeCP2 prefers to bind H3K27me3-modified nucleosome, and that the MeCP2 occupancy genome-wide are influenced by the status of H3K27me3.

**MeCP2-H3K27me3 interaction is independent of DNA methylation.** To further determine whether MeCP2 binding to H3K27me3 is independent of DNA methylation, we examined MeCP2-chromatin interactions in cells with minimal DNA methylation. DKO1 cells[28] contain DNMT1 and DNMT3b double deletions (Supplementary Fig. 8A). Chromatins of DKO1 cell have <10% of DNA methylation when compared to the parental cells, HCT116[11]. The MeCP2 enrichment comparison between HCT116 and DKO1 revealed that most of MeCP2 enriched loci remain unchanged ($n = 403$ K of 481 K, 83.7%). There are also de novo MeCP2 enriched loci appeared in DKO1 cells ($n = 100$ K of 503 K, 19.9%) (Supplementary Fig. 8B). MeCP2 enrichment changes were further analyzed in differential methylation regions (DMRs) between HCT116 and DKO1. We reanalyzed the WGBS data of both HCT116 and DKO1 from Blattler et al.[11], and identified 91,106 DMRs (Supplementary Fig. 8C). In DKO1 cells, there was on average 81.25% reduction in DNA methylation in DMRs when compared to HCT116 cells. Interestingly, MeCP2 enrichment was significantly increased in DMRs of DKO1 (Fig. 5a). These results suggested that DNA methylation is not the only determinant of MeCP2 chromatin interaction.

Next, to analyze MeCP2-H3K27me3 interaction in hypomethylated DNA, we analyzed the genome-wide correlations between H3K27me3 and MeCP2 enrichment by ChIP-seq and compared between HCT116 and DKO1. Consistent with neuronal tissue (Fig. 2g) and SH-SY5Y cells (Fig. 4b), MeCP2 enrichment is highly correlated with H3K27me3 in HCT116 cells ($r = 0.79$). With significant reduction of DNA methylation, in DKO1, we observed a strong correlation between H3K27me3 and MeCP2 binding as well ($r = 0.77$) (Fig. 5b). More specifically, changes in MeCP2 enrichment at the loci with increased or decreased H3K27me3 were compared (Fig. 5c–f). Consistent with previous studies[11,29], dynamic redistribution of H3K27me3 was observed throughout the genome of DKO1 compared to that of HCT116. Comparison between HCT116 and DKO1 showed that there was a large number of H3K27me3 enriched loci in DKO1 cells that were not detected in HCT116 ($n = 51$ K of 166 K, 30.7%); while only a small number of unique H3K27me3 loci were present in HCT116 ($n = 5$ K of 120 K, 4.4%) (Fig. 5c). This finding indicated that an increased H3K27me3 enrichment appeared in cells with decreased DNA methylation level. Among loci with increased H3K27me3 binding, 5000 loci were randomly selected (Fig. 5d). Analyzing MeCP2 enrichment within these loci, we observed an increase in MeCP2 binding in DKO1 (Fig. 5e). Changes of MeCP2 distribution in DKO1 were further visualized at selected regions (Fig. 5f). At chr 11: 129,230,000–129,250,000, DNA methylation was significantly depleted in DKO1 cells, while increased H3K27me3 binding was detected in this region. With input normalized ChIP-seq, we observed an increased MeCP2 enrichment within this region. The H3K27me3-dependent MeCP2 binding change was also observed in the H3K27me3 reduced loci in DKO1 (Supplementary Fig. 8D–F). Overall, these results further support that MeCP2 binds to H3K27me3 enriched loci independently of DNA methylation.

**MeCP2 binds to transcriptional regulatory loci with H3K27me3.** MeCP2-chromatin interaction in mouse OE was abundant among intronic and intergenic regions (Fig. 6a). The regulatory regions, 5′UTR and promoter-TSS, were less abundant with MeCP2 binding when only considering peak numbers.

However, the regulatory regions are under-represented in the mouse genome and intronic and intergenic regions are extensive in length. We statistically evaluated the association between MeCP2 binding frequencies to these regions by taking the structure of genome into account using regioneR[30], testing overlaps of MeCP2 enriched loci to genomic regions based on permutation sampling. MeCP2 was highly enriched in the 5′UTR, promoter-TSS, and exons by Z score (Z score > 60, Fig. 6b and Supplementary Fig. 9A-C). Conversely, although MeCP2 binding was the most abundant, by absolute count, in intergenic loci genome-wide, its association with each region was not significant based on Z score (Fig. 6b).

The distribution of MeCP2 with epigenetic modifications was further visualized at selected regions to explore the relationship between MeCP2 and epigenetic modifications. At chr 17: 45,680,000 – 45,700,000, which includes *Tmem151b* and *Nfkbie*, while example correlations were observed between MeCP2 binding and H3K27me3, co-enrichment of MeCP2 and H3K9ac is limited at TSS including 5′UTR (Fig. 6c, top three wiggle tracks). In addition, no distinct co-enrichment between MeCP2 binding and methylated DNA (CpG, CAG, and CAH) was observed (Fig. 6c, bottom three wiggle tracks). At genome-wide level, the co-enrichment of MeCP2 and H3K27me3 or H3K9ac was evaluated in the annotated regions. H3K27me3 peaks were present in 38.13% of MeCP2+/5′UTR regions, 32.48% of MeCP2+/exon regions, and 20.99% of MeCP2+/promoter-TSS regions, whereas comparably less H3K27me3+ enrichment was identified in MeCP2+/intergenic (8.57%) sites. Furthermore, the transcriptional regulatory region including MeCP2+/5′UTR (33.55%) and MeCP2+/promoter-TSS (15.22%) is relatively highly co-enriched with both H3K27me3 and H3K9ac (Fig. 6d). From the canonical point of view that MeCP2 binds to methylated CpG, this strong enrichment of MeCP2 in regions around TSS could be due to a higher level of DNA methylation. We further analyzed MeCP2 binding with the degree of DNA methylation around TSS regions (Fig. 6e). Contrary to the existing notion, the extended region of TSS showed a separation between DNA methylation and MeCP2 binding. The distribution of MeCP2 and histone PTMs, H3K27me3 and H3K9ac, were highly correlated (Fig. 6f). In addition, epigenetic modifications were analyzed in enhancer loci (Supplementary Fig. 9G), defined by co-enrichment of DNase I hypersensitive sites (DHS), H3K4me1 and H3K27ac[31,32]. Within these enhancer loci, MeCP2, H3K27me3, and H3K9ac were abundant (Fig. 6h); while DNA is hypomethylated (Fig. 6g). These results demonstrate that MeCP2 binding to gene expression regulatory regions is independent of DNA methylation but correlated with both H3K27me3 and H3K9ac.

**MeCP2-H3K27me3 cooperatively modulates gene expression.** The coexistence of histone PTMs and MeCP2 in transcription regulatory regions implies that they could influence the regulation of gene expression. To examine the relationship between gene expression and MeCP2 binding with histone PTMs in mouse OE, we evaluated RNA-seq data[13] after clustering genes depending on MeCP2 binding and histone PTMs distribution. A super self-organizing map (SuperSOM) clustering algorithm[33] was used to define the characteristics of co-occurrence of MeCP2 and specific histone PTMs. SuperSOM clustering for the region encompassing 1 kb upstream and 5 kb downstream sequences of all genes revealed three discrete clusters: Cluster1, co-enrichment of MeCP2 and H3K27me3 in the TSS and gene body (4812 genes); Cluster2, co-enrichment of MeCP2 and H3K27me3 in the TSS and gene body, and H3K9ac enrichment in TSS (770 genes); and Cluster3, co-enrichment of MeCP2, H3K27me3 and H3K9ac in

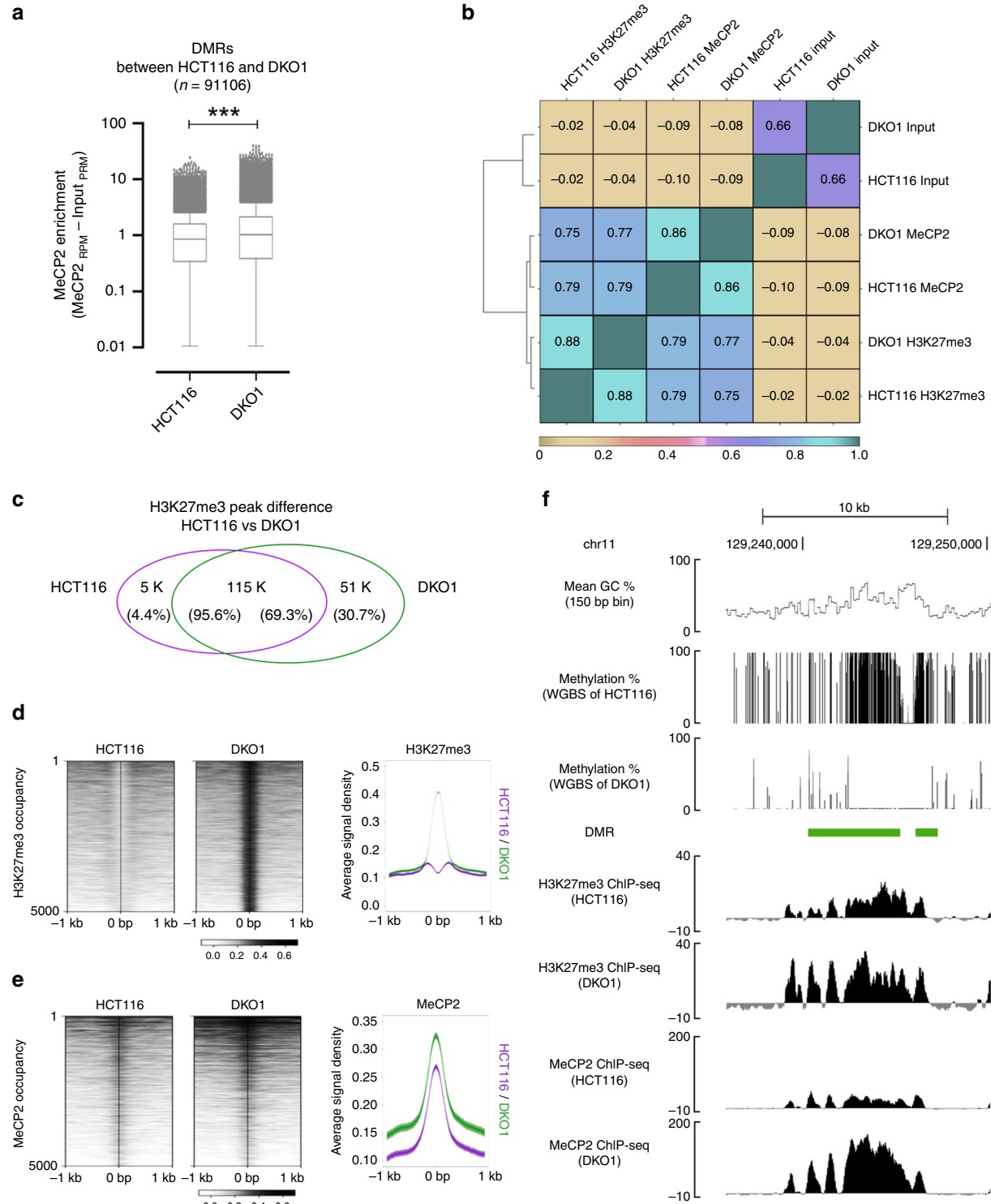

**Fig. 5 MeCP2 binds to H3K27me3 enriched loci independent of DNA methylation. a** MeCP2 enrichment at differential methylated regions (DMRs) between HCT116 and DKO1 cells. $N = 2$ (per genotype, HCT116 vs DKO1) from biologically independent cells. n = 91,106, $p < 0.0001$ (two-tailed Wilcoxon signed-rank test, ***$p < 0.0001$). Box-and-whisker plots show median, 10–90 percentile, and min and max values. p = 0.00 were reported as $p < 0.0001$. **b** Genome-wide inter-correlation among Input, H3K27me3 ChIP-seq, MeCP2 ChIP-seq in both HCT116 and DKO1 (binning size = 200 bp). Global Pearson correlation analyses between input, H3K27me3 and MeCP2 across HCT116 and DKO1. **c** Venn diagram showing the number of common and unique H3K27me3 peaks between HCT116 and DKO1. **d** Heatmap illustrating randomly selected 5000 loci with increased H3K27me3 in DKO1. Peaks are sorted according to signal intensity. Averaged H3K27me3 signals are given for HCT116 (purple) and DKO1 (green). **e** Heatmap showing changes in MeCP2 occupancy within corresponding H3K27me3 increased loci of DKO1 shown in **d**. Peaks are sorted according to signal intensity. Averaged MeCP2 signals are given for HCT116 (purple) and DKO1 (green). **f** Representative tracks of MeCP2 binding, DNA methylation and H3K27me3 enrichment in HCT116 and DKO1 cells at chr11: 129,230,000–129,250,000. UCSC genome-browser view (hg19) of the CpG density (per 150 bp) is shown on top. DMR is indicated in green bar. MeCP2 and H3K27me3 enrichment are input normalized. The shaded areas are up to ±S.D. from the average profile (**d** and **e**).

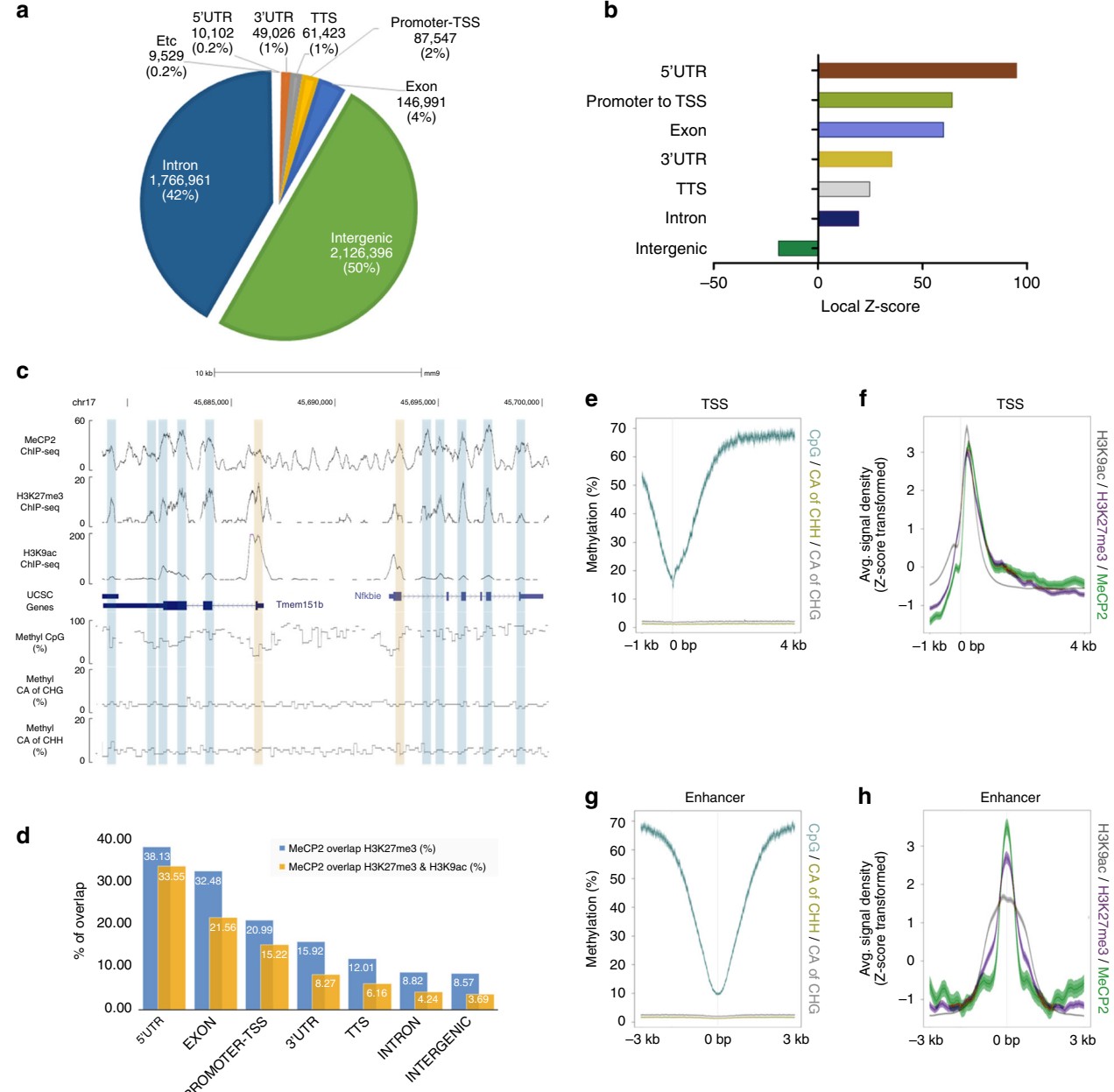

**Fig. 6 Coexistence of MeCP2, H3K27me3, and H3K9ac in transcription regulatory regions. a** Genome-wide distribution of MeCP2 binding loci, determined by PING. Mouse OE is the tissue source for all subsequent analyses in this figure. **b** MeCP2 binding frequency determined by Z-score in each annotated gene structural regions. **c** UCSC genome-browser view (mm9) of MeCP2, H3K27me3, H3K9ac ChIP-seq, and DNA methylation levels at a region on chr17: 45,675,956–45,700,322. The transparent blue bars indicate loci where MeCP2 and H3K27me3 coexist. The transparent orange bars point to the co-enrichment of MeCP2, H3K27me3, and H3K9ac. **d** Percentage of overlap between MeCP2 binding and H3K27me3 (blue) or H3K27me3 with H3K9ac (orange) within each annotated regions. **e** Methylation levels of CpG, CAG and CAH in 1 kb up- and 4 kb downstream of the TSS. **f** Z-score transformed average enrichment of MeCP2, H3K27me3, and H3K9ac, calculated 1 kb up- and 4 kb downstream of the TSS. **g** Methylation levels of CpG, CAG and CAH in 3 kb up- and 3 kb downstream of the enhancer loci (defined in Supplementary Fig. 9G). **h** Z-score transformed average enrichment of MeCP2, H3K27me3 and H3K9ac, calculated −3 kb to +3 kb around the midpoint of each enhancer (defined in Supplementary Fig. 9G). The shaded areas are up to ±S.D. from the average profile (**e-h**).

the TSS (2101 genes) (Fig. 7a). To determine whether the spatial positioning of MeCP2 with histone PTMs was related to differences in gene expression, we compared the levels of gene expression using three WT biological replicates[13] (Fig. 7b). RNA-Seq analysis of genes from cluster1 showed that these genes were expressed at minimum levels on average, whereas there was an appreciable amount of transcription in cluster2 (median RPKM [log10] of cluster1 = 0.26 and cluster2 = 0.97). Cluster3 (median

RPKM [log10] = 1.11) exhibited relatively strong transcriptional activity, compare to those of clusters1 and 2. Taken together, while H3K9ac enrichment around the TSS is indicative of higher expression levels in general, the presence of MeCP2 and H3K27me3 in the gene body on average correlates with lower transcriptional levels.

To further investigate the MeCP2 function in transcriptional regulation, we examined the alterations of gene expression within

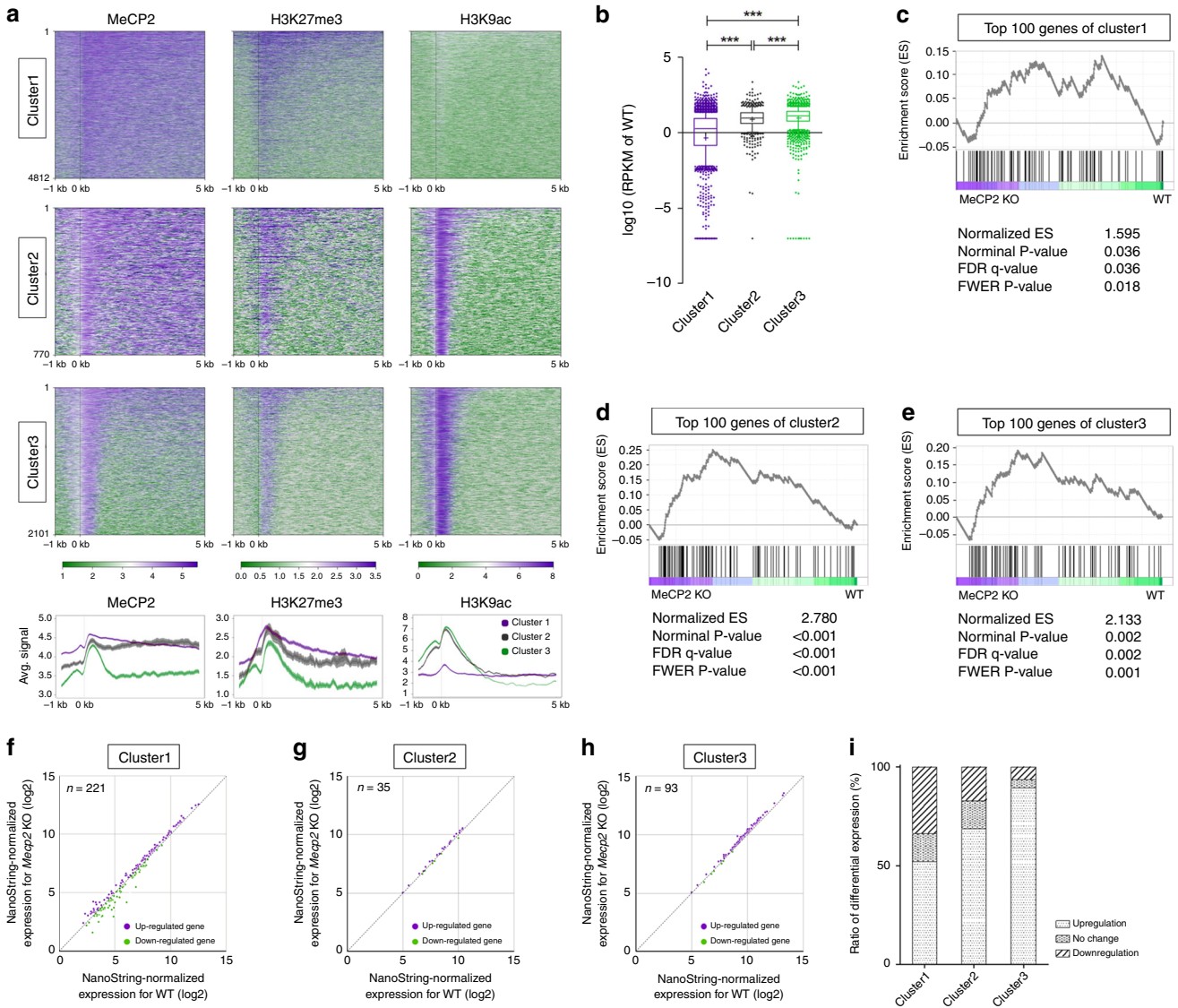

**Fig. 7 MeCP2 differentially regulates gene expression depending on H3K27me3 and H3K9ac pattern. a** Gene clusters classified by co-modification of MeCP2, H3K27me3, and H3K9ac within 1 kb up- and 5 kb downstream of the TSS. Genes sorted by descending order of mean signal intensity (numbers of genes in each cluster indicated on the left). Average profile of MeCP2 (bottom left), H3K27me3 (bottom center), and H3K9ac (bottom right). The shaded areas are up to ±S.D. from the average profile. Mouse OE is the tissue source for all subsequent analyses in this figure. **b** The expression levels of genes from cluster1 (purple, n = 4,159), cluster2 (gray, $n = 628$), and cluster3 (green, $n = 1747$). $N = 3$ (WT) from biologically independent animals. $p = 7.70 \times 10^{-73}$ (cluster1 vs cluster2); $p = 2.25 \times 10^{-218}$ (cluster1 vs group3); $p = 2.93 \times 10^{-5}$ (cluster2 vs cluster3). (two-tailed Mann–Whitney testing with Bonferroni corrections, ***$p < 0.0001$). Box-and-whisker plots show median, 10–90 percentile, and min and max values. Mean is marked by "+." **c–e** Differential gene expression between WT and *Mecp2* KO was shown by enrichment scores, determined by GSEA, within each cluster1 (**c**), cluster2 (**d**) and cluster3 (**e**). $N = 3$ (per genotype, WT vs *Mecp2* KO) from biologically independent animals. **f–h** Comparison of gene expression between WT and *Mecp2* KO using amplification-free NanoString technology. $N = 2$ (per genotype, WT vs *Mecp2* KO) from biologically independent animals. Purple dots indicate up-regulated genes, and green dots represent down-regulated genes in cluster1 (**f**), cluster2 (**g**) and cluster3 (**h**). **i** The percentage of differentially expressed genes in each cluster in the NanoString nCounter gene expression assay.

the above-defined clusters between WT and *Mecp2* knockout (KO) mice. Gene set enrichment analysis (GSEA)[34] for the top 100 genes in each group was performed (Fig. 7c–e). Though both up- and down-regulation of gene expression were observed within cluster1 genes when comparing WT to KO (normalized enrichment score (NES) = 1.595, Fig. 7c), up-regulated genes in cluster2 and 3 were more abundant than down-regulated genes comparing KO vs WT (NES = 2.78, Fig. 7d; NES = 2.133, Fig. 7e). In order to further confirm these observations, using an amplification-free gene expression analysis (NanoString), we compared the expression differences between WT and KO in

genes belonging to each cluster (Fig. 7f–i, Supplementary Table 4). Consistent with the GSEA analysis, the percentage of up-regulated genes in cluster2 (68%, Fig. 7g, i) is higher than those of cluster1 (52%, Fig. 7f, i). A high percentage of up-regulated genes was also measured in cluster3 (89%, Fig. 7h and i). The differential expression of genes belonging to clusters2 and 3 was re-confirmed using RT-qPCR (Supplementary Fig. 10B, C). These data indicate that MeCP2 suppresses genes modified with high levels of H3K9ac at the TSS.

To perform an unbiased analysis of MeCP2, H3K27me3, and H3K9ac occupancy in regions of differentially expressed genes in

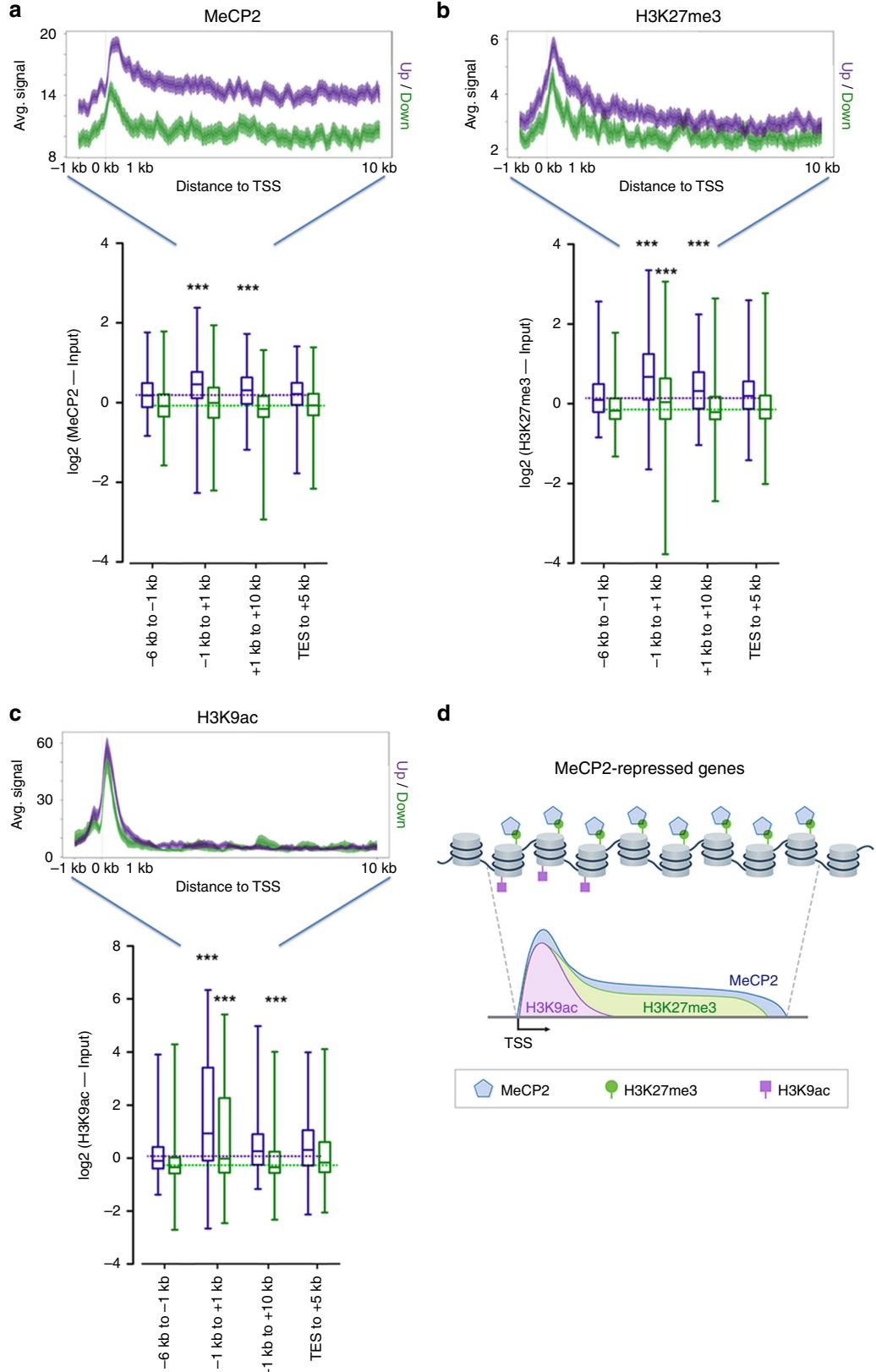

*Mecp2* KO mice, we isolated all of upregulated (FDR < 0.05, fold change > 1.3, $n = 890$) and down-regulated genes (FDR < 0.05, fold change < −1.3, $n = 699$). MeCP2 binding between 1 kb upstream and 10 kb downstream of each TSS was mapped (Fig. 8a). While MeCP2 binding is abundant among up-regulated genes, particularly at the TSS, the binding is substantially lower

within the corresponding region for down-regulated genes. To investigate the relative enrichment of MeCP2 in the intragenic region to that of flanking regions, we compared the average value of the MeCP2 enrichment signal across predefined regions (TSS regions, −1 to 1 kb and gene body, 1 to 10 kb) to that of the proximal regions of the gene body (−6 to −1 kb and transcription

**Fig. 8 Distribution of MeCP2, H3K27me3 and H3K9ac. a–c** Distribution of MeCP2, H3K27me3 and H3K9ac within differentially expressed genes between WT and *Mecp2* KO. $N = 3$ (per genotype, WT vs *Mecp2* KO) from biologically independent animals. MeCP2 (**a**), H3K27me3 (**b**), and H3K9ac (**c**) distribution in wild-type chromatin plotted 1 kb up- and 10 kb downstream of the TSS of genes that are up- (purple, $n = 890$) or down-regulated (green, $n = 699$) in *Mecp2* KO. $N = 2$ (WT) from biologically independent animals. For MeCP2 (**a**), $p = 6.62 \times 10^{-30}$ (Up, TSS vs proximal regions); $p = 4.13 \times 10^{-7}$ (Up, Gene body vs proximal regions); $p = 0.05$ (Down, TSS vs proximal regions); $p = 0.02$ (Down, Gene body vs proximal regions). For H3K27me3 (**b**), $p = 1.79 \times 10^{-43}$ (Up, TSS vs proximal regions); $p = 2.02 \times 10^{-6}$ (Up, Gene body vs proximal regions); $p = 2.53 \times 10^{-5}$ (Down, TSS vs proximal regions); $p = 0.26$ (Down, Gene body vs proximal regions). For H3K9ac (**c**), $p = 2.90 \times 10^{-32}$ (Up, TSS vs proximal regions); $p = 0.17$ (Up, Gene body vs proximal regions); $p = 1.88 \times 10^{-6}$ (Down, TSS vs proximal regions); $p = 3.21 \times 10^{-4}$ (Down, Gene body vs proximal regions). (two-tailed Mann–Whitney test, ***$p < 0.0001$). Box-and-whisker plots show median, 10–90 percentile, and min and max values. The shaded areas are up to ±S.D. from the average profile (**a–c**). **d** Schematic diagram of MeCP2 chromatin interaction and its function in gene expression regulation.

ending site [TES] to +5 kb). In the up-regulated genes, MeCP2 enrichment is significantly higher at the TSS and gene body (median MeCP2 [log2] for TSS = 0.45, gene body = 0.31 and proximal regions = 0.19), whereas MeCP2 was evenly distributed among different genomic regions in the down-regulated genes.

H3K27me3 enrichment within the up-regulated genes was also markedly higher than in down-regulated genes (Fig. 8b). When comparing KO vs WT, we also observed in up-regulated genes that H3K27me3 enrichment was significantly higher in the TSS and gene body than in any of the distal regions (median H3K27me3 [log2] for TSS = 0.67, gene body = 0.31, and proximal regions = 0.14). In contrast, H3K27me3 enrichment was restricted to the proximal region of the TSS in down-regulated genes (median H3K27me3 [log2] for TSS = 0.04, gene body = −0.21, and proximal regions = −0.16). Overall the H3K27me3 enrichment of both up- and down-regulated genes closely paralleled to the extent and magnitude of MeCP2 enrichment (Fig. 8a, b). Contrary to MeCP2 and H3K27me3, remarkably high enrichment of H3K9ac at the TSS was observed in both up- and down-regulated genes (Fig. 8c). Strong H3K9ac enrichment around the TSS, with little concentration throughout the body of the gene, was observed in both up- and down-regulated genes (median H3K9ac [log2] for TSS = 0.93, gene body = 0.26, and proximal regions = 0.04 in up-regulated genes; median H3K9ac [log2] for TSS = −0.01, gene body = −0.35, and proximal regions = −0.29 in down-regulated genes). Therefore, in addition to H3K9ac at the TSS, co-occurrence of MeCP2 and H3K27me3 at the TSS and gene body modulate gene expression (Fig. 8d).

## Discussion

MeCP2 has been defined as a methylated DNA-binding protein[2,8,9]. However, there are many exceptions in vivo as to the specificity of MeCP2 binding to methyl-cytosine. Particularly, maintaining MeCP2 bindings in chromatin in *Dnmt1*, *Dnmt3a*, *Dnmt3b* triple knock out ES cells argue that MeCP2 interacts with chromatin via additional mechanisms[35]. In the present study, we show that MeCP2 directly binds to the nucleosome complex and its binding is enhanced by the presence of modified H3, H3K27me3. Furthermore, MeCP2 interactions with H3K27me3 correlate with its role in transcriptional repression.

Under in vitro conditions, it was reported that the C-terminal region of MeCP2 contributes to the formation of complexes with nucleosomes and that the binding was enhanced with methylated DNA[36,37]. An independent study using small angle X-ray scattering technique[38] confirmed that MeCP2 binds near the nucleosomal dyad, close to the linker DNA entry-exit region. Furthermore, genome-wide analyses of MeCP2 binding patterns have shown that MeCP2 enrichment co-occurred at nucleosome positions[8] or histone H1 binding sites[17]. Recently, we also demonstrated that MeCP2 preferentially localizes at GC-rich regions and co-localizes with nucleosomes in a genome-wide study[13]. However, the direct interaction between MeCP2 and

nucleosomal protein and the effect of histone modifications on MeCP2 binding to chromatin has not been well studied.

In this study, we provide evidence that MeCP2 physically associates with nucleosomal proteins and co-localizes with mononucleosomes genome-wide (Fig. 1). Specifically, MeCP2 binding is highly correlated with H3K27me3 distribution (Fig. 2). In vitro pull down and immunoprecipitation assay reveals that MeCP2 directly binds to histone H3, and this binding is enhanced by H3K27me3 (Fig. 3). Manipulation of H3K27me3 levels affects genome-wide MeCP2 binding (Fig. 4), suggesting that H3K27me3 is an important factor in determining the recruitment of MeCP2 to specific loci. We also found that MeCP2 binding to chromatin was maintained in hypomethylated cell lines as well as that the alteration of MeCP2 binding correlated with the change in H3K27me3 (Fig. 5). The MBD of MeCP2 has been characterized as a binding region for 5-methylcytosine (mC)[39] by Bird and coworkers, and subsequent studies have shown that MeCP2 also has a high affinity to non-CG methylation (mCH)[8,9] and 5-hydroxy methyl cytosine (hmC)[10]. We performed domain-specific characterization on MeCP2's affinity to H3K27me3 and observed higher affinity of MBD to H3K27me3 containing nucleosomes compared to unmodified nucleosomes (Fig. 3b–g). Considering that the MBD of MeCP2 binds to both methylated (or hydroxylmethylated) cytosine and H3K27me3, the two binding motifs may be competitive to MeCP2 binding, depending on the epigenetic status of the locus. Indeed, it was observed that MeCP2 binding is enriched with H3K27me3 in transcription regulatory regions (TRR) where DNA is hypomethylated (Fig. 6e–h). On the other hand, the enrichment of MeCP2 relative to H3K27me3 levels in the gene body was not as obvious as TRR (Fig. 7a), which possibly means that MeCP2 enrichment is also mediated by other factors such as methylated DNA. In general, considering that DNA is hypermethylated in gene body except for TRR (Fig. 6e), MeCP2 binding in the gene body is likely due to methylated DNA as well as H3K27me3.

Previously, we reported that mCpG% at TSSs was not significantly different between the up- and down-regulated genes, whereas MeCP2 enrichment at TSSs of up-regulated genes is higher than that of the down-regulated genes in *Mecp2* KO[13]. In this study, we demonstrate that H3K27me3 at TSSs is significantly higher in up-regulated genes compared to that of the down-regulated genes of *Mecp2* KO, and MeCP2 binding for chromatin is also proportional to the level of H3K27me3 in both up and down-regulated transcripts (Fig. 8a, b).

It is well established that H3K27me3 is associated with transcription repression[40,41]. The deposition of H3K27me3 is aggravated when transcription is in an inactive state[42]. It is possible that H3K27me3 enrichment recruits MeCP2 and the H3K27me3 deposition together with MeCP2 is responsible for transcription repression. Therefore, the upregulated genes identified in *Mecp2* KO mice may indicate a partial alleviation of transcriptional repression by MeCP2. However, not all genes marked by both MeCP2 and H3K27me3 are upregulated in *Mecp2* KO mice. The

level of acetylation in H3K9 around the TSS also contributes to gene expression alterations in *Mecp2* KO mice (Figs. 6 and 7).

MeCP2 has the ability to bind nucleosomal proteins as shown in Fig. 1b, c. Other histones and their modifications could also contribute to MeCP2 binding to chromatin. For example, MeCP2 competes with histone H1 for chromatin binding[43] and that histone H1.2, a subfamily of histone H1, has binding affinity for H3K27me3[44]. Therefore, it is possible that MeCP2 could compete with histone H1 for binding to H3K27me3. Also, a large proportion of chromatin is modified by more than one epigenetic modification, which suggests that there are complex parameters for controlling the MeCP2 binding affinity to the whole-genome and for gene expression regulation. Therefore, it is necessary to study the interaction with other epigenetic modulations in correlation with MeCP2 to achieve better understanding of MeCP2 function in the regulation of gene expression.

## Methods

**Animals.** The MeCP2 null strain was generated by A. Bird and colleagues[4]. These mutant mice (B6.129P2(C)-Mecp2tm1.1Bird/J, stock number 003890) were acquired from Jackson Laboratories (Bar Harbor, Maine). Control animals were WT littermates of the mutant mice. All procedures are approved by the University of California, Davis animal Care and Use Committee and in accordance with the NIH animal use guidelines and institution-approved animal use protocols.

**Cell culture.** SH-SY5Y cell line (Korean Cell Line Bank, 22266) were maintained in DMEM/F12 (Gibco, 11320033) supplemented with 10% fetal bovine serum (Welgene, S001-01) containing Penicillin/Streptomycin (Lonza, 17-602E) at 5% CO$_2$ in 37 °C. HCT116 cell line and DNMT1 (Δexons3-5/Δexons3-5); DNMT3B (−/−) of HCT116 (horizon, HD R02-079), named as DKO1, were maintained in RPMI 1640 (Sigma, R8758), supplemented with 10% FBS (Welgene, S001-01) containing Penicillin/Streptomycin (Lonza, 17-602E) at 5% CO$_2$ in 37 °C. 293T cell line were maintained in DMEM (Welgene, LM 001-05) supplemented with 10% FBS(Welgene, S001-01) containing Penicillin/Streptomycin (Lonza, 17-602E) at 5% CO$_2$ in 37 °C. Drosophila S2 cells (ATCC, CRL-1963) were cultured in Schneider's Drosophila Medium (Gibco, 21720024) supplemented with 10% FBS (Welgene, S001-01) in 25 °C.

**Immunoprecipitation assays.** To extract nuclei, olfactory epithelium (OE) tissue were isolated and physically dissociated in PBS. The dissociated cells, which are mostly olfactory sensory neurons (OSNs) are spun down 1000×g at 4 °C for 10 min, and lysed with lysis buffer (5 mM PIPES, 85 mM KCl, 0.5% IGEPAL CA-630, pH 8.0) with Halt protease and phosphatase inhibitor cocktail (Thermo Scientific, 1861281). Plasma membranes were mechanically disrupted by a homogenizer (Kimble-Kontes. 885300-0002), while the nucleus remains intact after homogenization. Nuclei were pelleted at 2400×g for 10 min at 4 °C, resuspend the nuclei pellet in 300 μl of Ab binding and washing buffer (NaCl 150 mM, Tris 50 mM, NP-40 1%, pH 8.0) and sonicated by Bioruptor UCD200 (Diagenode) for 5 min (30 s/ 30 s cycle). 125 U of Benzonase Nuclease (EMD chemical, 70746-3) were added to each sample to digest associated DNA (Supplementary Fig. 1B). and the protein preparations were saved for Co-IP assay. For Co-IP, 4.8 μg of MeCP2 antibody (Diagenode, pAb-052-050) or 8 μg of pan histone H1 antibody (Santa cruz, sc-34464) or 4.8 μg of rabbit IgG (Millipore, 12-370) as negative control were added to 50 μl of Dynabeads Protein A/G (Invitrogen) and incubated with nuclear extract for 30 min at room temperature. The pull-down complex was boiled in 2x SDS buffer to elute protein for SDS-PAGE gel, and visualized by Coomassie staining.

SH-SY5Y cells were treated with either DMSO or 5 μM GSK343 for 72 h. After washing with PBS, SH-SY5Y cells were lysed in IP150 lysis buffer containing 50 mM Tris-HCl (pH 7.6), 150 mM NaCl, 0.5% (v/v) NP-40, 1 mM EDTA, 10% Glycerol and protease/phosphatase inhibitors (1 mM Na$_2$VO$_4$, 10 mM NaF, 2 mM PMSF, 5 μg/ml Lepeptin, 10 μg/ml Aprotinin, 1 μg/ml Pepstain A). For Co-IP, 5 μg of histone H3 antibody (Abcam, ab1791), 5 μg of MeCP2 antibody (Diagenode, pAb-052-050), or 5 μg of rabbit IgG (Diagenode, kch-504-250) as negative control were added to 50 μl of protein G-Sepharose 4 Fast Flow (GE Healthcare) and incubated with the cell lysate for overnight at 4 °C. Beads were washed five times with IP150 lysis buffer and the pull-down complex was boiled in 2x SDS buffer to elute protein for SDS-PAGE gel. Western blotting was performed as per standard protocol.

**ChIP-seq in olfactory neuroepithelia.** Chromatin immunoprecipitation assays were performed using MAGnify chromatin immunoprecipitation system (Invitrogen, 49-2024). Specifically, olfactory neuroepithelia were dissected from the nasal cavity and dissociated mechanically via trituration in phosphate buffer saline (PBS). Cross-linking was done by incubating MOE cells with 1% formaldehyde for 5 min. After washing off the formaldehyde, MOE cells were lysed briefly in SDS

containing buffer followed by sonication for 15 min with 30 s intervals to shear genomic DNA using Bioruptor UCD200 (Diagenode). The sheared DNA was evaluated by electrophoresis for their size ranging between 200 and 300 bp. Each sheared genomic DNA preparation was split into two equal portions and incubated with either 50 μg of MeCP2 (Diagenode, pAb-052-050) or 15 μg of histone H1 (Santa Cruz, sc-34464) or 10 μg of H3K27me3 (Diagenode, pAb-069-050) or 10 μg of H3K9ac (Diagenode, pAb-177-050). After pulling down magnetically, genomic DNA in the complex was reverse cross-linked by DNase-free Proteinase K and purified. For ChIP-seq experiments, histone H1, H3K27me3, and H3K9ac ChIPed or input DNA were used to prepare the ChIP-seq library as described (NEXTflex ChIP-seq kit, Bio Scientific, NOVA 5143-01 and NEXTflex ChIP-seq Barcodes-6, NOVA 514120) followed by high throughput sequencing with Illumina Hi-seq 2000.

For ChIP-qPCR experiment, each sheared genomic DNA preparation was split into two equal portions and incubated with either 2 μg of MeCP2 (Diagenode, pAb-052-050) or 2 μg of histone H1 (Santa Cruz, sc-34464) or Rabbit IgG (Millipore, 12-370) as negative control. To examine MeCP2 binding at *Bdnf* locus, quantitative real-time PCR was performed on the precipitated DNA fragments using five pairs of oligonucleotide primers designed to produce amplicons covering the MeCP2 binding site in the *Bdnf* gene and flanking sequences (chr2: 109,514,240-109,517,676, NCBI37/mm9) (for details on primers positions, see Supplementary Fig. 3A and Supplementary Table 5). For Re-ChIP experiments, 25 μl of ReChIP buffer (Dilution Buffer, 10 mM DTT) was added to MeCP2-genomic DNA complex following washes and incubated at 37 °C for 30 min. The sample was then diluted 40 times in dilution buffer (Invitrogen, P/N 100006377), and then 2 μg of histone H1 (Santa Cruz, sc-34464) or 2 μg of rabbit IgG (Millipore, 12-370) for the second IP were incubated for 4 h at 4 °C. After genomic DNA in the complex was purified, quantitative real-time PCR was performed on the precipitated DNA fragments using two pairs of oligonucleotide primers in flanking regions of *Bdnf* gene (for details on primers information, see Supplementary Table 5).

**Plasmid constructs and cloning.** Human MeCP2 cDNA [NM_004992] was amplified from the HEK293T cells by RT-PCR using the following primers: forward, 5′-AAAACTCGAGGTAGCTGGGATGTTAGGGCTCA-3′ and reverse, 5′-AAAAGCGGCCGCTCAGCTAACTCTCTCGGTCACG-3′ and cloned into the pCI-neo-Flag mammalian expression vectors (Promega, Madison, WI, USA). To prepare the serial deletion constructs of MeCP2 (HMGD1/MBD [1-164aa], MBD/HMGD2 [75-207aa], TRD/CTD [207-486aa], CTD [310-486aa]), each fragment was PCR-amplified using pCI-neo-Flag-MeCP2 full-length as template, and the PCR products were inserted into the XhoI and NotI sites of pCI-neo-Flag vector (all amino acid positions were based on the sequence of accession NM_004992). For in vitro GST-Pulldown assay, MeCP2 full-length and MBD domain [75~164aa] of MeCP2 was cloned into pGEX4T-1 (GE Healthcare, Fairfield, CT, USA). All constructs were verified by sequencing. A comprehensive list of all PCR primers used in this study can be found in Supplementary Table 7.

**Pull-down assays.** For in vitro biotin-pulldown assays regarding to Fig. 3a, 20 pmol of GST-tagged MeCP2 (Abnova, H00004204-P01) were combined with either 20 pmol of recombinant biotinylated mononucleosomes (Active Motif, 31467) or 20 pmol of biotinylated mononucleosomes containing H3K27me3 (Active Motif, 81135) in a final volume of 100 μl binding Buffer (50 mM Tris-HCl pH 8.0, 0.1 mM EDTA, 1 mM DTT, 10% glycerol, 1 μl of Halt$^{TM}$ Protease Inhibitor (Thermo Fisher, 78430)), and incubated at 25 °C for 2 h. In all, 50 μl of Dynabeads M-280 Streptavidin (Thermo Fisher, 11205D) were used to immobilize biotinylated mononucleosomes at 25 °C for 30 min. After binding buffer washes, mono-nucleosome complexes were pulled-down and boiled in 2x SDS buffer to elute protein for SDS-PAGE gel for Western blot analyses.

For Flag-pulldown assays regarding to Fig. 3c, plasmid constructs transfected 293T cell lysates were mixed with 3 μg of anti-Flag antibody (Sigma, F1804) and incubated overnight at 4 °C in IP150 lysis buffer containing 50 mM Tris-HCl (pH 7.6), 150 mM NaCl, 0.5% (v/v) NP-40, 1 mM EDTA, 10% Glycerol and protease/phosphatase inhibitors (1 mM Na$_2$VO$_4$, 10 mM NaF, 2 mM PMSF, 5 μg/ml Lepeptin, 10 μg/ml Aprotinin, 1 μg/ml Pepstain A). Immune complexes were precipitated using protein G-Sepharose 4 Fast Flow (GE Healthcare, 17061801) at 4 °C for overnight and then washed five times with IP150 lysis buffer. The immunoprecipitated complex was boiled in 2x SDS buffer to elute protein for SDS-PAGE gel, and analyzed by Western blotting.

For in vitro GST-pulldown assay in Fig. 3d and Supplementary Fig. 5A, 3 μg of GST-tagged MeCP2 full-length (1-486aa) or MBD domain (75-164aa) was immobilized onto Glutathione Sepharose 4B beads (GE Heathcare) and subsequently incubated with 1.5 μg of recombinant biotinylated mononucleosome (Active Motif, 31467) or biotinylated mononucleosome containing H3K27me3 (Active Motif, 81135) overnight at 4 °C. After washing with lysis buffer (20 mM Hepes (pH 7.6), 150 mM NaCl, 5 mM MgCl$_2$, 1% Triton x-100 and 5% Glycerol), mononucleosome complexes were analyzed by Western blotting. H3K27 peptide (Activemotif, # 81048), or H3K27me3 peptide (Activemotif, # 81052) were used for peptides competition in Fig. 3e and Supplementary Fig. 5B.

**Western blotting**. Western blotting was performed as per standard protocol. The antibodies used were as follows: rabbit anti-MeCP2 (Cell signaling, D4F3), 1:1000; Chicken anti-MeCP2 (custom antibody, gift from Dr J.M. LaSalle), 1:1000; goat anti-histone H1 antibody (Santa cruz, sc-34464), 1:1000; rabbit anti-Histone H3 (Cell signaling, D1H2), 1:3000; mouse anti-Histone H4 (abcam, ab31830), 1:500; VeriBlot for IP secondary antibody (abcam, ab131366), 1:4000; Rabbit anti-Chicken HRP (IgY H&L) (abcam, ab6753), 1:50000; rabbit anti-Ezh2 (Cell signaling, D2C9), 1:2000; rabbit anti-H3K27me3 (Cell signaling, C36B11), 1:2000; rabbit anti-GAPDH (Cell signaling, D16H11) 1:3000; mouse anti-GST (Santa Cruz, sc-138) 1:200; rabbit anti-DNMT1 (Cell signaling, D63A6) 1:2000; rabbit anti-DNMT3B (Cell signaling, E8A8A) 1:1000; mouse anti-actin (Santa cruz, sc-47778) 1:1000.

**Immunofluorescent staining**. SH-SY5Y cells were grown on sterilized coverslip and treated with DMSO or 5 μM GSK343 (Sigma-Aldrich, SML0766) in DMSO for 72 h. The DMSO or GSK343-treated SH-SY5Y cells were fixed with 4% paraformaldehyde (PFA). Primary antibodies used in this study include: rabbit anti-H3K27me3 (Cell signaling, C36B11), 1:1000; mouse anti-MeCP2 (Sigma-Aldrich, Men-8). Nuclei were visualized with DAPI (Invitrogen, 00-4959-52). Images were taken using a Nikon Eclipse 80i fluorescence microscope and compiled using Adobe Photoshop CS5.

**Quantification of LINE-1 methylation**. LINE-1 methylation levels were quantified using a pyrosequencing assay at EpigenDx (Worcester, MA). Briefly, 500 ng of purified gDNA was bisulfate treated and purified using Zymo DNA Methylation Kit (Zymo research, Orange, CA). LINE-1 regions were subsequently amplified using DNA polymerase. Using Pyrosequencing PSQ96 HS System (Pyrosequencing Qiagen), a T/C SNP was analyzed individually at each of 4 loci using QCpG software (Pyrosequencing Qiagen). Degree of methylation was individually calculated as C% = C/(C + T) at each of 4 loci. Average of methylation status at all four loci were reported as overall percent 5meC status.

**Targeted bisulfite sequencing**. Targeted Bisulfite-Seq library were prepared using the SureSelect$^{XT}$ Methyl-Seq Target Enrichment System for Illumina Multiplexed Sequencing (Agilent Technologies). Briefly, 3 μg of genomic DNA per sample was fragmented to median size of 150 bp, repaired, and ligated to adapters. The prepared gDNA library was hybridized using SureSelectXT methyl-Seq capture reagent, followed by bisulfite conversion using EZ DNA Methylation-Gold kit (ZymoResearch, D5005). After desulphonation, the sequence-modified, target-enriched library were amplified and uniquely indexed by PCR. The indexed libraries were pooled and multiplexed into 4 separate lanes for 100 bp paired-end sequencing (Illumina HiSeq2500).

**ChIP with reference exogenous genome (ChIP-Rx)**. SH-SY5Y cells (density of $0.5-0.6 \times 10^6$ cells/ml) treated with DMSO or 5 μM GSK343 for 72 h were washed with 10 ml of PBS and cross-linked with 1% formaldehyde for 5 min. The cross-linking reactions were quenched with 10x Glycine stop solution. The cross-linked SH-SY5Y cells were lysed in lysis buffer (5 mM PIPES, 85 mM KCl, 0.5% IGEPAL CA-630, pH 8.0) using a dounce homogenizer (Kimble-Kontes, 885300-0002) to aid in nuclei release. The released nuclei were pelleted by centrifugation (10 min, $2300 \times g$ at 4 °C) and stored at −80 °C. As a reference exogenous genome, Drosophila S2 cells were cross-linked with 1% formaldehyde for 5 min and were quenched with 10× Glycine stop solution. The Drosophila S2 cells were then washed three times with ice cold PBS. Washed cell pellets were flash frozen and stored at −80 °C. For each ChIP-Rx experiment, SH-SY5Y cells and Drosophila S2 cells ratio of 3:1, 12 million cross-linked SH-SY5Y nuclei and 4 million crosslinked S2 cells, were combined and sonicated for 15 min with 30 s intervals to shear genomic DNA using Bioruptor XL (Diagenode). The sheared genomic DNA was evaluated by electrophoresis for their size ranging between 200 bp and 300 bp. Each sheared genomic DNA preparation was incubated with 10 μg of H3K27me3 (Diagenode, pAb-069-050) or 20 μg of MeCP2 (Diagenode, pAb-052-050). After pulldown, genomic DNA in the complex was reverse cross-linked by DNase-free Proteinase K and purified. For ChIP-seq experiments, H3K27me3 or MeCP2 ChIPed, or input DNA were used to prepare ChIP-seq libraries as described (NEXTflex ChIP-seq kit, Bio Scientific, NOVA 5143-01 and NEXTflex ChIP-seq Barcodes-6, NOVA 514120) followed by high throughput sequencing with Illumina NextSeq500. To determine H3K27me3 PTM or MeCP2 binding to human *HOXB* loci, quantitative real-time PCR was performed using three pairs of oligonucleotide primers covering *HOXB* cluster loci after ChIP (for details on primers information, see Supplementary Table 5).

HCT116 or DKO1 cells (density of $0.5-0.6 \times 10^6$ cells/ml) were cross-linked with 1% formaldehyde for 5 min. The crosslinking reactions were quenched with Glycine stop solution. The cross-linked HCT116 or DKO1 cells were lysed in lysis buffer (5 mM PIPES, 85 mM KCl, 0.5% IGEPAL CA-630, pH 8.0) using a dounce homogenizer (Kimble-Kontes. 885300-0002) to aid in nuclei release. The released nuclei were pelleted by centrifugation (10 min, $2300 \times g$ at 4 °C) and stored at −80 °C. As a reference exogenous genome, Drosophila S2 cells were cross-linked in parallel. For each ChIP-Rx experiment, HCT116 or DKO1 and Drosophila S2 cell lysates were combined with ratio of 3:1 and sonicated for 15 min with 30 s intervals

to shear genomic DNA using Bioruptor XL (Diagenode). The sheared genomic DNA was evaluated by electrophoresis for targeted size ranging between 200 and 300 bp. Each sheared genomic DNA preparation was incubated with 10 μg of H3K27me3 (Diagenode, pAb-069-050) or 20 μg of MeCP2 antibody (Diagenode, pAb-052-050). After pulldown, genomic DNA in the complex was reverse cross-linked by DNase-free Proteinase K and purified. For ChIP-seq experiments, H3K27me3 or MeCP2 ChIPed, or input DNA were used to prepare ChIP-seq libraries as described (NEXTflex ChIP-seq kit, Bio Scientific, NOVA 5143-01 and NEXTflex ChIP-seq Barcodes-6, NOVA 514120) followed by high throughput sequencing with Illumina NextSeq500.

**Quantitative RT-PCR**. Olfactory epithelial tissue from three wild-type and three *Mecp2* KO mice were dissected. Total RNAs were extracted using TRIZOL reagent (Invitrogen). cDNAs were obtained by reverse transcription (RT) using Superscript II reverse transcriptase (Invitrogen, 18064014). For endogenous control gene selection, transcription levels of 16 commonly used housekeeping genes were measured using TaqMan Array Mouse Endogenous Controls Plate (ThermoFisher Scientific, 4426699). For gene expression change validation between WT and *Mecp2* KO, quantitative RT-PCR experiments were done in triplicate using Bio-Rad CFX96 Real-time PCR detection system.

SH-SY5Y cells treated with DMSO or 5 μM GSK343 for 72 h were washed with 10 ml of PBS and total RNAs were extracted using TRIZOL reagent (Invitrogen). For *MECP2* gene expression change analysis between DMSO and GSK343-treated cell, cDNAs were obtained by QuantiTect Reverse Transcription (Qiagen, 205311) and quantified by Luna (NEB, M3003L) using Bio-Rad CFX96 Real-time PCR detection system (for details on primers information, see Supplementary Table 6). Relative expressions were determined by ΔΔCt.

**ChIP-seq and MNase-seq data analysis (mouse OE)**. The histone H1 (GSM3465057 and 3465058), H3K9ac (GSM3465059 and 3465060), and H3K27me3 (GSM3465061 and 3465062) ChIP-seq reads were aligned to the mm9 reference genome using Bowtie 2[45] with the options --sensitive --score-min L,−1.5, −0.3. Also, the published Input[13] (GSM1827604 and 1827605), MeCP2 ChIP-seq reads[13] (GSM1827606 and 1827607), and MNase-seq reads[13] (GSM1827608) were aligned under the same protocol (Supplementary Table 1). PCR duplicates were removed using samtools rmdup for further analysis. The unique aligned reads were then expanded to an experimentally determined fragment length of 200 bp, but the unique alignment reading of the MNase-seq data were extended to a maximum of 146 bp for calculation of nucleosome occupancy.

**Bisulfite sequencing analysis (mouse OE)**. Published whole-genome bisulfite sequencing reads[13] (GSM1827616, and 1827617) were aligned to the mm9 reference genome using BSseeker2[46]. Total 271 million reads were aligned and methylation levels were estimated using BSseeker2 (bs_seeker2-call_methylation.py) with default parameters.

**RNA-seq data analysis (mouse OE)**. Published RNA-seq reads[13] (GSM1827610, 1827611, and 1827612 from WT, and GSM1827613, 1827614, and 1827615 from KO) were aligned to the mm9 refGene[47] using TopHat 2[48]. In the WT biological replicates, readings of 65, 55 and 56 million were uniquely mapped and correctly paired. Under the same conditions, 58, 51, and 51 million reads were obtained in *Mecp2* KO biological replicates. For differential gene expression analysis on refGene transcripts, cufflinks and cuffdiff[49] were used with default parameters.

**H3K27me3 and MeCP2 ChIP-seq data analysis (SH-SY5Y)**. The H3K27me3 ChIP-Rx (GSM3465067 and 3465068 from DMSO-treated sample, and GSM3465069 and 3465070 from GSK343-treated sample), the MeCP2 ChIP-Rx (GSM3489380 and 3489381 from DMSO-treated sample, and GSM3489382 and 3489383 from GSK343-treated sample) and Input (GSM3465063 and 3465064 from DMSO-treated sample, and GSM3465065 and 3465066 from GSK343-treated sample) libraries generated from DMSO or GSK343-treated SH-SY5Y cells were sequenced in multiplex on an NextSeq500 using 75 bp single end sequencing. The sequences (75-nt tags, single end) were separately aligned to either Homo sapiens (hg19) or D. melanogaster genome (dm3) using Bowtie 2[45] with the options --sensitive --score-min L,−1.5,−0.3 (Supplementary Table 1). PCR duplicate were removed for further analysis. For quantitative comparisons of H3K27me3 ChIPed reads between the samples, the exogenous genome derived normalization factor (α-x) for each experiment was determined according to the protocol of Orlando et al[50] as follows: 1 over the number of reads mapping to D. melanogaster genome (dm3) per million. Then, the uniquely aligned readings for Homo sapiens (hg19) were standardized using the normalization factor, α-x, for quantitative comparisons between samples (for detailed information, see Supplementary Table 3). For quantitative comparisons of MeCP2 ChIPed reads between the samples, the normalization factor (α) for each experiment was derived as follows: 1 over the number of reads mapping to Homo sapiens (hg19) per million. Then, the uniquely aligned readings for Homo sapiens (hg19) were standardized using the normalization factor, α, for quantitative comparisons between samples (for detailed information, see Supplementary Table 3).

**H3K27me3 and MeCP2 ChIP-seq data analysis (HCT116 and DKO1)**. The H3K27me3 ChIP-Rx (GSM4041346 and 4041347 from HCT116, and GSM4041348 and 4041349 from DKO1), the MeCP2 ChIP-Rx (GSM4041350 and 4041351 from HCT116, and GSM4041352 and 4041353 from DKO1) and Input (GSM4041342 and 4041343 from HCT116, and GSM4041344 and 4041345 from DKO1) libraries were sequenced in multiplex on a NextSeq500 using 75 bp single end sequencing. The sequences (75-nt tags, single end) were separately aligned to either major chromosome (1 to 22, X and Y chromosome) of Homo sapiens (hg19) using Bowtie2[45] with the options --sensitive --score-min L,−1.5,−0.3 (Supplementary Table 1). PCR duplicate were removed for further analysis.

**Targeted bisulfite sequencing analysis (SH-SY5Y cells)**. The targeted bisulfite sequencing library (GSM4041354 and 4041355 from DMSO-treated sample, and GSM4041356 and 4041357 from GSK343-treated sample) were aligned to Homo sapiens (hg19) using Bowtie2[45] (Supplementary Table 2). After deduplication (deduplicate_bismark), methylation levels (bismark_methylation_extractor) were determined using bismark tools[51] with default parameters.

**NanoString gene expression analysis (mouse OE)**. Transcriptional levels of WT (GSM3465071 and 3465072) or Mecp2 KO (GSM3465073 and 3465074) OE tissues were measured using pre-made sets of panel, nCounter Mouse Neuropathy Panel Kit (XT-CSO-MNROP1-12, NanoString Technologies) containing 770 genes. The measurements were performed in biological duplicates. The nSolver Analysis Software 4.0 (NanoString Technologies) were used for data analysis.

**Statistics and reproducibility**. Regarding to ChIP-seq analyses, extended experimental procedures are included in Supplementary Method section of Supplementary Information. All ChIP-seq assays were performed two times with high reproducibility (Supplementary Fig. 2A–M). All immunoprecipitation assays were performed at least three times (Fig. 1a–c; Fig. 4a; Supplementary Fig. 5C). All pull-down assays were performed at least two times (Fig. 3a, c–e; Supplementary Fig. 5A, B). All results were successfully repeated. Following experiments were performed once (Supplementary Figs. 1A, B, 6A, and 8A).

**Reporting summary**. Further information on research design is available in the Nature Research Reporting Summary linked to this article.

## Data availability

Sequencing data have been deposited into NCBI Gene Expression Omnibus under accession code GSE71126, and GSE122366. The source data underlying Figs. 1b, c, 3a, c–e, 4a and Supplementary Figs. 1A, 6A and 8A are provided as a Source data file. All data is available from the corresponding author upon reasonable request. Source data are provided with this paper.

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

## Acknowledgements

This research was supported by the National Research Foundation (NRF) funded by the Korean government (NRF-2016R1D1A1A02937023, NRF-2019R1I1A3A01062768 to W. L., and NRF- 2020R1A2C2007845, MRC-2015-009070 to T.O.) and NIH R01DC011346 to Q.G.. We thank Dr. Jang Jae Lee for statistical analyses, Dr. Sangsoon Woo for instruction of PING program, Ramesh Mariappan for technical contributions, and Dr. Richard P. Tucker and Dr. Younghoon Kee for critical reading of the paper.

## Author contributions

W.L., T.O, and Q.G. designed the experiments, W.L., J.Y., and J.K. performed the experiments and analyzed the data. W.L., T.O., and Q.G. prepared the manuscript. All authors read and approved the final paper.

## Competing interests

The authors declare no competing interests.
