## [Peer Review File · Nature Communications]

Reviewers' Comments:

Reviewer #1:

Remarks to the Author:

The study by Lee et al mainly describes histone H3K27me3 and histone H1 profiling in mouse olfactory epithelium cells. These are correlated with RNA-Seq analysis of genes affected by a small molecule inhibitor of EZH2, an enzyme establishing H3K27me3, and with their previous analysis of a chromatin regulator MeCP2 ChIP-seq and RNA-seq of genes regulated by MeCP2. The authors report a significant co-localization between MeCP2, H3K27me3 and histone H1 and conclude that MeCP2 regulates genes by binding to H3K27me3. Their ChIP-seq data are of good quality and may provide a valuable resource for those studying role of MeCP2 in gene regulation. Still, the results reported here are largely descriptive and do not provide sufficient mechanistic insights for direct interaction between MeCP2 and H3K27me3, especially in view that all these three markers are widely distributed across the genome notable positive correlation ($r > 0.5$) with the general distribution of the nucleosomes released by MNase-seq. To provide more meaningful mechanistic analysis, direct and specific interaction of MeCP2 with nucleosomes containing H3K27me3 histone or its competition with H3K27me3 peptides (see Lewis, et al., Science, 2013. 340(6134): p. 857-61. for example) remains to be conducted. In addition, these observations are not quite novel since partial co-localization between MeCP2 and histone H1 had been shown before (cited in ref. 16) and association between histone H1 and H3K27me3 had been also shown in an important work of Kim et al (Scientific reports, 2015. 5: p. 16714) not cited by Lee et al..

My other, more specific, concerns are as follows:

- On page 6, the subtitle: "MeCP2 has high affinity to methylated histones as well as methylated cytosine" is not correct since MeCP2 co-localization and not affinity with H3K27me3 are reported in the manuscript.
- On page 12, regarding the statement: " MeCP2 and H3K27me3 co-occurrence at the TSS and gene body suppresses gene expression (Fig. 5D)." - the causative role of MeCP2 and H3K27me3 co-occurrence is not sufficiently demonstrated by experiments.
- Similarly, on page 13, the statement: "the co-enrichment of MeCP2 and H3K27me3 peaks around the TSS, at which H3K27me3 and DNA methylation are mutually exclusive, clearly demonstrates that H3K27me3 is a specific substrate for MeCP2 binding (Fig. 3E)." is way too strong as the co-enrichment does not demonstrate a specific binding of H3K27me3 by MeCP2.
- On Fig. 1 A and C, the authors show co-IP with histones including histone H3. Western blotting with antibodies against histone H3K27me3 should be conducted to monitor association with this histone modification.
- On Fig. 2A, they show very strong peaks of MNase-seq correlated with equally strong MeCP2-enriched genomic peaks. These data present a significant caveat since MNase produces different peaks at different extent of digestion. It appears that certain enrichment of MeCP2 on some loci and absence at the others may be due to different degrees of chromatin degradation by MNase. MNase digestion profiles at several extents of digestion (Mieczkowski, et al., Nature comm., 2016. 7: p. 11485) need to be mapped and compared to MeCP2 to determine robust enrichment in MeCP2 over the input.

Reviewer #2:

Remarks to the Author:

The study by Lee et al. demonstrates that a significant portion of MeCP2 binding to chromatin is mediated by its physical association with the nucleosome complexes containing H3K27me3 modification. Although MeCP2 has been well known for its binding affinity to methylated DNA, genome-wide evidence was presented for MeCP2 co-localization with H3K27me3-enriched nucleosomes devoid of noticeable DNA methylation. Such binding mechanism of MeCP2 was further shown to be significant in gene regulation. Overall, the presented data reasonably supports the conclusion and clearly shows the overlap between MeCP2 and nucleosomes enriched with H3K27me3 at gene promoter regions. Previous studies already implicated the association of MeCP2 with repressive histone modification markers such as H3K27me3 and H3K9me3, and there has been an intricate relationship reported between DNA methylation and histone modification profiles. The significance of the current study would be the genome-wide demonstration of MeCP2 co-localization with H3K27me3-enriched nucleosomes which could potentially underlie the MeCP2 function in transcriptional regulation. However, the evidence described in the current manuscript does not yet convincingly prove the functional significance of such mechanism in transcriptional control especially with regard to the relative contributions from DNA methylation and histone modifications to MeCP2 recruitment, and additional analysis would be necessary before considering its publication in Nature Communications. Below are the detailed comments/questions.

1. In order to properly examine the functional significance of histone modification-dependent and DNA methylation-independent chromatin binding of MeCP2 in transcriptional regulation, all relevant analysis should be paralleled with comparing DNA methylation profiles. For example, in Fig. 5, the authors' model could be better supported if DNA methylation profiles in up- and down-regulated genes are analyzed in comparison. One issue is that, due to the intricate interdependent relationship between histone modifications and DNA methylation, it might be difficult to assess the functional consequence of H3K27me3-dependent MeCP2 association without influencing DNA methylation profiles. In other words, it is not clear if the current study convincingly demonstrates the sole impact of histone modification-dependent recruitment of MeCP2 in transcriptional regulation.
2. One way to further support the authors' model would be to alter DNA methylation levels and examine if MeCP2 binding profiles at H3K27me3-enriched nucleosomes would be influenced with transcription changes although it might be beyond the scope of the current study.
3. The enrichment analysis of histone modifications and MeCP2 in Fig. 2D should consider the fact that the levels of mononucleosomes are different between the groups, and therefore, for the histone modifications, the normalization of enrichment values to the levels of mononucleosomes within each group would be necessary.
4. Histone acetylation is generally considered to be associated with gene activation whereas H3K27me3 and MeCP2 are enriched at transcriptionally repressed genes. Many MeCP2-bound sites exhibit high levels of H3K9ac, and the best-correlative marker with transcription levels shown in Fig. 4 is H3K9ac profiles at TSSes regardless of MeCP2/H3K27me3 bindings. This implies that the binding level of MeCP2 and H3K27me3 per se can't explain or be indicative of the transcription outcome.
5. Fig. 4 seems to show that transcription levels are inversely correlated with MeCP2 enrichment over the gene body areas, but not much with the enrichment levels at TSSes (especially as shown in Fig. 4A). H3K27me3 shows less obvious enrichment over the gene body areas. However, the most clearly demonstrated co-localization of MeCP2 with H3K27me3-enriched nucleosomes independently of DNA methylation is seen at TSSes as shown in Fig. 3E. The authors need to discuss this in details.
6. In Fig. 3, despite more than 90% of MeCP2 binds to intronic and intergenic regions, the authors claim that only 5'UTR, promoters, and exons exhibit the significant binding of MeCP2 according to the analysis using regioneR. First, The authors should explain how regioneR works with the rationale. Secondly, there must be numerous enhancers present in intergenic and intronic regions, which are the critical gene regulatory regions for transcription. However, the authors' analysis didn't consider this at all and instead lump all the intergenic or intronic regions as a whole for the enrichment analysis. The same is true for the analysis of MeCP2 distribution with epigenetic markers.

7. It would be informative to test if MeCP2 can directly interact with H3K27me3-enriched nucleosomes. Fig. 1 shows an association of MeCP2 with nucleosome complexes, not necessarily proving whether the interaction is direct or not. Although MeCP2 does not have to be in direct contact with nucleosomes in the authors' model, it might provide more supporting evidence for it.
8. Fig. 3E nicely shows tight colocalization of MeCP2 with histone modifications at TSSes that are devoid of CpG methylation, but the profile of tri-nucleotides methylation sites at TSSes should also be examined.
9. The specificity of MeCP2 antibody should be demonstrated.
10. Some figure labels should be larger. (ex. Fig. 1D and E, Fig. 3C and D).

Reviewer #3:

Remarks to the Author:

IP-MS/Western data alone are enough to prove physical interaction in this specific case, because MeCP2 is closely associated with chromatin/nucleosome. The fact that MeCP2 pulls down H1, H3 and H4 at the same time suggest the nucleosome came down together, since it is impossible for MeCP2 to physically bind/touch all three histones at the same time. To demonstrate that MeCP2 physically binds to H1, H3, H4 separately, it is necessary to map out the domain on MeCP2 essential for binding to each of the histones. Such results could provide mechanistic insights when combined with how Rett syndrome causing mutations can disrupt each of the binding event.

The study failed to provide direct evidence that MeCP2 physically binds to H3K27me3 (as depicted in Fig 5D). Given MeCP2's known association with DNA and chromatin, it is not surprising to observe co-localization between MeCP2 and certain histone modifications in some parts of the genome. A more important question is how MeCP2 is recruited to each locus in response to what signals. Although the study describes many correlations between MeCP2 binding, histone modification, nucleosome location, and gene expression, no novel mechanistic insights are provided. Results from the drug treatment are not adequate because of potential side effects and indirect actions.

Overall, the current study represents only incremental gain of knowledge as compared to what the same lab and others have previously reported (see line 60-63 on page 4).

Author response to reviewers

Title: MeCP2 regulates gene expression through recognition of H3K27me3

Authors: Wooje Lee, Jung-mi Yun, Jeeho Kim, Takbum Ohn, Qizhi Gong

Below we address each reviewer's comments/concerns in a point-by-point format.

Underline denotes reviewers' comments, *Red font* denotes edited words, *yellow shade* denotes highlighted words

- manuscript entitled "MeCP2 regulates gene expression through recognition of H3K27me3" has now been seen by 3 referees. You will see from their comments below that while they find your work of interest, some important points are raised. We are interested in the possibility of publishing your study in Nature Communications, but would like to consider your response to these concerns in the form of a revised manuscript before we make a final decision on publication. **We consider it particularly important that the revised manuscript to provide test of direct interaction between MeCP2 and H3K27me3-enriched nucleosome, as well as the other referee comments.**

We therefore invite you to revise and resubmit your manuscript, taking into account the points raised. Please highlight all changes in the manuscript text file.

We are committed to providing a fair and constructive peer-review process. Do not hesitate to contact us if you wish to discuss the revision in more detail or if there are specific requests from the reviewers that you believe are technically impossible or unlikely to yield a meaningful outcome.

Reviewers' comments:

Reviewer #1 (Remarks to the Author):

The study by Lee et al mainly describes histone H3K27me3 and histone H1 profiling in mouse olfactory epithelium cells. These are correlated with RNA-Seq analysis of genes affected by a small molecule inhibitor of EZH2, an enzyme establishing H3K27me3, and with their previous analysis of a chromatin regulator MeCP2 ChIP-seq and RNA-seq of genes regulated by MeCP2.

The authors report a significant co-localization between MeCP2, H3K27me3 and histone H1 and conclude that MeCP2 regulates genes by binding to H3K27me3. Their ChIP-seq data are of good quality and may provide a valuable resource for those studying role of MeCP2 in gene regulation. Still, the results reported here are largely descriptive and do not provide sufficient mechanistic insights for direct interaction between MeCP2 and H3K27me3, especially in view that all these three markers are widely distributed across the genome notable positive correlation ($r > 0.5$) with the general distribution of the nucleosomes released by MNase-seq. To provide more meaningful mechanistic analysis, direct and specific interaction of MeCP2 with nucleosomes containing H3K27me3 histone or its competition with H3K27me3 peptides (see Lewis, et al., Science, 2013. 340(6134): p. 857-61. for example) remains to be conducted. In addition, these observations are not quite novel since partial co-localization between MeCP2 and histone H1 had been shown before (cited in ref. 16) and association between histone H1 and H3K27me3 had been also shown in an important work of Kim et al (Scientific reports, 2015. 5: p. 16714) not cited by Lee et al.

Response: We appreciate the thoughtful comments from Reviewer #1 which improved our manuscript. As the reviewer pointed out, we agree that it is important to demonstrate the direct interaction between MeCP2 and H3K27me3. To address the reviewer's concern (to strengthen the conclusion of genome-wide colocalization between MeCP2 and H3K27me3), we have performed the following set of new experiments:

- (1) First, as reviewer 1 suggested, we performed pull-down assay using either non-modified mono-nucleosome or mono-nucleosomes containing H3K27me3 for GST-tagged MeCP2 *in vitro* condition. We show MeCP2 more strongly interact with mono-nucleosomes containing H3K27me3 rather than non-modified mono-nucleosome *in vitro* condition (Fig. 3A).
- (2) Second, we performed *in vivo* immunoprecipitation assay with cell lysates from either GSK343 - or DMSO treated SHSY5Y cells. Compared to the DMSO control, MeCP2 co-precipitation with histone H3 was greatly reduced in GSK343 treated sample suggesting an importance of H3K27me3 in association with MeCP2. (Fig. 3E).
- (3) Third, we analyzed the effect of H3K27me3 levels on MeCP2 binding to chromatin in genome-wide. We demonstrated that a decrease in H3K27me3 induces a decrease in MeCP2 binding to chromatin. (Fig. 3F and G).

We believe that the *in vitro* and *in vivo* biochemical assays, and genome-wide analysis of MeCP2 redistribution by pharmacological treatment provide a compelling evidence.

Also, we add the reference (Scientific reports, 2015. 5: p. 16714) in discussion (shown as follows).

(Line 451 of page 16, current revision) “A previous study showed that MeCP2 competes with histone H1 for chromatin binding⁴⁰ and that histone H1.2, a subfamily of histone H1, has binding affinity for H3K27me3⁴¹.”

My other, more specific, concerns are as follows:

- On page 6, the subtitle: “MeCP2 has high affinity to methylated histones as well as methylated cytosine” is not correct since MeCP2 co-localization and not affinity with H3K27me3 are reported in the manuscript.

Response: We thank Reviewer #1 for pointing this out. We agree that it does not include the evidence that MeCP2 has a high affinity for H3K27me3 **until Figure 2**. We have now corrected the wording in the text (shown as follows).

(Line 127 of page 6, original manuscript and Line 123 of page 6, current revision) Subtitle: ‘MeCP2 ~~has high affinity to methylated histones~~ **highly co-localized with H3K27me3** as well as methylated cytosine.’

- On page 12, regarding the statement: “MeCP2 and H3K27me3 co-occurrence at the TSS and gene body suppresses gene expression (Fig. 5D).” the causative role of MeCP2 and H3K27me3 co-occurrence is not sufficiently demonstrated by experiments.

- Similarly, on page 13, the statement: “the co-enrichment of MeCP2 and H3K27me3 peaks around the TSS, at which H3K27me3 and DNA methylation are mutually exclusive, clearly demonstrates that H3K27me3 is a specific substrate for MeCP2 binding (Fig. 3E).” is way too strong as the co-enrichment does not demonstrate a specific binding of H3K27me3 by MeCP2.

Response: We do agree and thank Reviewer #1 for pointing this out. As we stated above, we now include *in vitro* and *in vivo* data that MeCP2 has a higher binding affinity for H3K27me3 than unmodified mononucleosomes in Figure 3. Also, we included new data to demonstrate MeCP2 bind to H3K27me3 independently of DNA methylation. We believe that the newly added data in this revised manuscript support the idea that H3K27me3 is a specific substrate for MeCP2 binding. In addition, to avoid any ambiguity in the text, we edited the sentence as shown below.

(Line 378 - 381 of page 13, original manuscript and Line 461 of page 16, current revision) ~~“the co-enrichment of MeCP2 and H3K27me3 peaks around the TSS, at which H3K27me3 and DNA methylation are mutually exclusive, clearly demonstrates that H3K27me3 is a specific substrate for MeCP2 binding (Fig. 3E).~~ **These results suggest that H3K27me3 is an important factor in determining the recruitment of MeCP2 to specific loci independent of DNA methylation (Fig. 4E and 4F).”**

Regarding to the previous comment,

(Line 346 - 347 of page 12, original manuscript and Line 426 of page 15, current revision) ~~“MeCP2 and H3K27me3 co-occurrence at the TSS and gene body suppresses gene expression (Fig. 5D)~~ **Therefore, in addition to H3K9ac at the TSS, co-occurrence of MeCP2 and H3K27me3 at the TSS and gene body modulate gene expression (Fig. 6I).**” We would like keep the sentence, because we provide the additional evidence that H3K27me3 is specific substrate for MeCP2 binding.

- On Fig. 1 A and C, the authors show co-IP with histones including histone H3. Western blotting with antibodies against histone H3K27me3 should be conducted to monitor association with this histone modification.

Response: As requested, we added a new Western blot in supplementary fig. 5C in which MeCP2 has an affinity to H3K27me3 in OE tissue. We also added a sentence in the text as shown below.

(Line 223 of page 9, current revision) “Also, we found that the pull-down proteins from OE tissue by MeCP2 antibody (Fig. 1C) contain H3K27me3 (Supplementary Fig. 5C).”

- On Fig. 2A, they show very strong peaks of MNase-seq correlated with equally strong MeCP2-enriched genomic peaks. These data present a significant caveat since MNase produces different peaks at different extent of digestion. It appears that certain enrichment of MeCP2 on some loci and absence at the others may be due to different degrees of chromatin degradation by MNase. MNase digestion profiles at several extents of digestion (Mieczkowski, et al., Nature comm., 2016. 7: p. 11485) need to be mapped and compared to MeCP2 to determine robust enrichment in MeCP2 over the input.

Response: We thank the reviewer for pointing this out. To optimize MNase digestion for sequencing experiment, we tested MNase digestion profiles of olfactory neuroepithelia chromatin at three different concentrations.

We observed a decrease of multiple nucleosome and an increase of mono nucleosomes as the concentration of MNase treatment increased (Fig.1. A) as also shown in Mieczkowski et al (2016) study. The MNase-seq libraries were prepared from the medium (0.5 Unit) and highest MNase digestion (5 Unit)

in the parallel experiment. The MNase-seq reads of high MNase concentration (MNase-hiC) is compatible with the number of reads from median MNase concentration treated MNase-seq (MNase-medC). In the MNase-hiC, we observed new peaks that did not appear in MNase-medC (Fig.1.B, loci with shadow). These peaks seem to correspond to heterochromatin as suggested in Mieczkowski *et al.* study. The exclusive peaks, not observed in MNase-medC, might be useful for a future study to reveal the relationship between MeCP2 and epigenetic status of chromatin.

Regarding the concept of the present study, we found a disadvantage of the MNase-hiC condition. Based on a quality control analysis using **ChIP-seq Analytic and Confidence Estimation (CHANCE)** (Diaz *et al.*, 2012), we found that while the signal strength of the MNase-hiC is not significant (Divergence test q-value, qFDR = 0.255; red line in Fig.1C), the signal strength of the MNase-medC is significant (Divergence test q-value, qFDR = 0.0465: sky blue in Fig.1C). Moreover, because we characterized epigenetic modification of MeCP2 non-binding nucleosomes, MeCP2-binding nucleosome, and nucleosome free MeCP2 binding loci **at single nucleosome unit**, a high-quality library with low background is required to define the mononucleosome. We were unable to define a single nucleosome unit in Mnase-hiC due to a reduction of signal to noise. We do understand Reviewer #1's concern, however, we believe that there is a limit to the MeCP2 enrichment comparative analysis in the MNase digestion profile depending on the degree of digestion in our study. In the canonical point of view, the quality of the current MNase-seq data, MNase-medC, is qualified and optimized for following reasons: 1) A good signal strength in quality control analysis (Fig.1.C, sky blue), 2) A typical occupancy pattern were observed in combined analysis with RNA-seq (Fig1. D).

Reviewer #2 (Remarks to the Author):

The study by Lee et al. demonstrates that a significant portion of MeCP2 binding to chromatin is mediated by its physical association with the nucleosome complexes containing H3K27me3 modification. Although MeCP2 has been well known for its binding affinity to methylated DNA, genome-wide evidence was presented for MeCP2 co-localization with H3K27me3-enriched nucleosomes devoid of noticeable DNA methylation. Such binding mechanism of MeCP2 was further shown to be significant in gene regulation. Overall, the presented data reasonably supports the conclusion and clearly shows the overlap between MeCP2 and nucleosomes enriched with H3K27me3 at gene promoter regions. Previous studies already implicated the association of MeCP2 with repressive histone modification markers such as H3K27me3 and H3K9me3, and there has been an intricate relationship reported between DNA methylation and histone modification profiles. The significance of the current study would be the genome-wide demonstration of MeCP2 co-localization with H3K27me3-enriched nucleosomes which could potentially underlie the MeCP2 function in transcriptional regulation. However, the evidence described in the current manuscript does not yet convincingly prove the functional significance of such mechanism in transcriptional control especially with regard to the relative contributions from DNA methylation and histone modifications to MeCP2 recruitment, and additional analysis would be necessary before considering its publication in Nature Communications. Below are the detailed comments/questions.

In order to properly examine the functional significance of histone modification-dependent and DNA methylation-independent chromatin binding of MeCP2 in transcriptional regulation, all relevant analysis should be paralleled with comparing DNA methylation profiles. For example, in Fig. 5, the authors' model could be better supported if DNA methylation profiles in up- and down-regulated genes are analyzed in comparison.

Response: We appreciate valuable suggestions to improve our manuscript. Since the previous study (Rube et al., 2016) included the data set relevant to this issue, we added the reference in the discussion section with following sentence.

Fig. 2. meCpG% profiles in either up- and down-regulated genes of *Mecp2* KO
The plot shows the meCpG% profiles (colour) in 8 kb regions surrounding the significantly up- and down-regulated genes after *Mecp2* KO (separated by thick horizontal black line). The TSSs are ordered by fold change in expression between KO and WT. The bottom plot shows the mean meCpG% in up- and down-regulated genes.

(Page 470, line 16) “Previously, we reported that mCpG% at TSSs was not significantly different between the up- and down-regulated genes, whereas MeCP2 enrichment at TSSs of up-regulated genes is higher than that of the down-regulated genes in *Mecp2* KO¹².”

2. One issue is that, due to the intricate interdependent relationship between histone modifications and DNA methylation, it might be difficult to assess the functional consequence of H3K27me3-dependent MeCP2 association without influencing DNA methylation profiles. In other words, it is not clear if the current study convincingly demonstrates the sole impact of histone modification-dependent recruitment of MeCP2 in transcriptional regulation.

One way to further support the authors' model would be to alter DNA methylation levels and examine if MeCP2 binding profiles at H3K27me3-enriched nucleosomes would be influenced with transcription changes although it might be beyond the scope of the current study.

Response: We appreciate this thoughtful suggestion and agree with a potential importance for this study. We now included a comprehensive analysis of the MeCP2 distribution for H3K27me3 in DKO1 cells showing extensive loss of DNA methylation (10% of the overall DNA methylation levels relative to the parental HCT116 cell line). MeCP2 turned out to be redistributed depending on increasing or decreasing H3K27me3 in DKO1 cell lines. The result is now included in Fig. 4 and Page 10. We do not seek further analysis regarding in gene expression caused by MeCP2 in the cell line, because it is beyond the scope of the current study.

3. The enrichment analysis of histone modifications and MeCP2 in Fig. 2D should consider the fact that the levels of mononucleosomes are different between the groups, and therefore, for the histone modifications, the normalization of enrichment values to the levels of mononucleosomes within each group would be necessary.

Response: Due to technical difference in genomic DNA processing for MNase-seq and ChIP-seq, it is challenging to directly normalize histone epigenetic modification signals to nucleosome levels. Therefore, Heatmap and aggregation plot of MeCP2, H3K27me3 and H3K9ac ChIP-seq are displayed alongside those of MNase-seq to illustrate the levels of nucleosomes in each case. Current data show that while MeCP2-absent mononucleosome loci correlates with depleted H3K27me3 and H3K9ac (Group1), MeCP2-enriched mononucleosome loci show high levels of H3K27me3 with or without the presence of stable mononucleosomes (groups 2 and 3). We edited following text with figure rearrangement (Fig. 2D).

(Line 164-167 of page 7, original manuscript and Line 160 of page 7, current revision) “We observed that, ~~whereas group1 contains low levels of H3K27me3 and H3K9ac, group2 occupied regions are enriched with both H3K27me3 and H3K9ac. However, while H3K27me3 in group3 is comparable to that in group2,~~ H3K9ac in group3 is higher than that of group2 (Fig. 2D bottom). **with mononucleosomes clearly present, MeCP2-absent (group 1) and MeCP2-enrichment (group 2) were correlated with the levels of H3K27me3 and H3K9ac. At the same time, in the loci that nucleosomes are unstable (group 3), MeCP2 enrichment also showed strong correlation with H3K27me3. These results indicate that MeCP2 binding is correlated with H3K27me3 in both closed and open chromatin.**”

The physical association between MeCP2 and H3K27me3 is further supported by data obtained from additional experiments (Fig. 3 and 4)

4. Histone acetylation is generally considered to be associated with gene activation whereas H3K27me3 and MeCP2 are enriched at transcriptionally repressed genes. Many MeCP2-bound sites exhibit high levels of H3K9ac, and the best-correlative marker with transcription levels shown in Fig. 4 is H3K9ac profiles at TSSes regardless of MeCP2/H3K27me3 bindings. This implies that the binding level of MeCP2 and H3K27me3 per se can't explain or be indicative of the transcription outcome.

Response: We agree with the reviewer on this comment and edited the manuscript to reflect this opinion.

(Line 288-290 of page 11, original manuscript and Line 366 of page 13, current revision) “Taken together, while H3K9ac enrichment around the TSS is ~~correlated with~~ **indicative of** higher expression levels in general, the presence of MeCP2 and H3K27me3 in the gene body ~~exerts impact on transcription suppression~~ **on average correlates with lower transcriptional levels.**”

5. Fig. 4 seems to show that transcription levels are inversely correlated with MeCP2 enrichment over the gene body areas, but not much with the enrichment levels at TSSes (especially as shown in Fig. 4A). H3K27me3 shows less obvious enrichment over the gene body areas. However, the most clearly demonstrated co-localization of MeCP2 with H3K27me3-enriched nucleosomes independently of DNA methylation is seen at TSSes as shown in Fig. 3E. The authors need to discuss this in details.

Response: We thank the reviewer for pointing this out. Methylation of cytosine at the genome level is well characterized. They are more frequently observed in the gene body compared to the transcriptional regulatory region (TRR). MeCP2's affinity to DNA methylation has also been shown. Here, we infer that MeCP2 binding to the gene body, except for TRR, is contributed by its interaction with both H3K27me3 and methylated DNA. Discussion of these findings are added:

(Line 463 of page 16, current revision) “**Indeed, it was observed that MeCP2 binding is enriched in transcription regulatory regions (TRR) where DNA is hypomethylated (Fig. 5E to 5H). However, the enrichment of MeCP2 and H3K27me3 in the gene body was not as obvious as TRR (Fig. 6A), which means that MeCP2 enrichment is also mediated by other factors such as methylated DNA. In general, considering that DNA is hypermethylated in gene body except for TRR (Fig. 5E), MeCP2 binding in the gene body is likely due to methylated DNA as well as H3K27me3.**”

6. In Fig. 3, despite more than 90% of MeCP2 binds to intronic and intergenic regions, the authors claim that only 5'UTR, promoters, and exons exhibit the significant binding of MeCP2 according to the analysis using regioneR. **First**, The authors should explain how regioneR works with the rationale. **Secondly**, there must be numerous enhancers present in intergenic and intronic regions, which are the critical gene regulatory regions for transcription. However, the authors' analysis didn't consider this at all and instead lump all the intergenic or intronic regions as a whole for the enrichment analysis. The same is true for the analysis of MeCP2 distribution with epigenetic markers.

Response: Regarding to the **first comment**, regioneR iteratively samples random regions in the genome considering matching size and chromosomal distribution of the experimentally-derived genomic regions. Then test overlaps of the experimentally-derived genomic regions to the genomic regions **based on permutation sampling**. This strategy allowed us to statistically evaluate whether MeCP2 peaks coincide with functional features, as shown in Fig. 5B. In addition, the MeCP2 peaks shifted in either direction of functional features to demonstrate the association of MeCP2 was strictly dependent on the exact location of the peak in Supplementary Fig. 9A to 9F. As suggested, we now include a rationale for regioneR in the main text as written below:

(Line 191-193 of page 8, original manuscript and Line 312 of page 12, current revision) ‘We statistically evaluated the association between MeCP2 binding frequencies to these regions by taking the structure of genome into account using regioneR, ~~for the association analysis of genomic regions~~ **testing overlaps of MeCP2 enriched loci to genomic regions based on permutation sampling.**’

Regarding to the **second comment**, Original Figure 3. A-E is a simple analytical concept derived from a structure based on genomic equivalence. We analyzed the MeCP2 enrichment and histone PTM on flanking regions of OE specific enhancer and provided the data in Figure 5 with description in the main text as written below:

(Line 337 of page 12, current revision) **“In addition, epigenetic modifications were analyzed in enhancer loci (Supplementary Fig. 9G), defined by co-enrichment of DNase I hypersensitive sites (DHS), H3K4me1 and H3K27ac^{32,33}. Within these enhancer loci, MeCP2, H3K27me3, and H3K9ac were abundant (Fig. 5H); however, DNA is hypomethylated (Fig. 5G). These results demonstrate that MeCP2 binding to gene expression regulatory regions is independent of DNA methylation but correlated with both H3K27me3 and H3K9ac.”**

7. It would be informative to **test if MeCP2 can directly interact with H3K27me3-enriched nucleosomes**. Fig. 1 shows an association of MeCP2 with nucleosome complexes, not necessarily proving whether the interaction is direct or not. Although MeCP2 does not have to be in direct contact with nucleosomes in the authors' model, it might provide more supporting evidence for it.

Response: We agree that this is a critical issue which will make our story more solid. We have now included additional data (Fig. 3A and 3E) to support that MeCP2 directly binds to H3K27me3 containing nucleosomes. Please also see our response to Reviewer 1.

8. Fig. 3E nicely shows tight colocalization of MeCP2 with histone modifications at TSSes that are devoid of CpG methylation, but the profile of tri-nucleotides methylation sites at TSSes should also be examined.

Response: We agree with the reviewer that such information will be useful for understanding comprehensive epigenetic mode for MeCP2 bindings. We have included the profile of tri-nucleotides

methylation sites at 1kb up- and 4kb downstream of the TSS in Fig. 5F including a description in the main text as written below.

(Line 333 of page 12, current revision) “We further analyzed MeCP2 binding with the degree of DNA methylation around TSS regions (Fig. 5E). Contrary to the existing notion, the extended region of TSS showed a separation between DNA methylation and MeCP2 binding.”

9. The specificity of MeCP2 antibody should be demonstrated.

Response: We added new data demonstrating the specificity of MeCP2 antibody used for co-immunoprecipitation assay and ChIP-seq library prep in Supplementary Fig. 1A. Also, we have previously reported the specificity of the MeCP2 antibody as supplementary data in Lee et al. (2014) and Rube et al. (2016).

10. Some figure labels should be larger. (ex. Fig. 1D and E, Fig. 3C and D).

Response: We have corrected those Figure labels.

Reviewer #3 (Remarks to the Author):

IP-MS/Western data alone are enough to prove physical interaction in this specific case, because MeCP2 is closely associated with chromatin/nucleosome. The fact that MeCP2 pulls down H1, H3 and H4 at the same time suggest the nucleosome came down together, since it is impossible for MeCP2 to physically bind/touch all three histones at the same time. To demonstrate that MeCP2 physically binds to H1, H3, H4 separately, it is necessary to map out the domain on MeCP2 essential for binding to each of the histones. Such results could provide mechanistic insights when combined with how Rett syndrome causing mutations can disrupt each of the binding event.

Response: We agree that identifying the domain of MeCP2 for interacting with Histone proteins is required for expanding the mechanical insight of MeCP2. However, the main focus of this study is to investigate the physical and functional relationships between MeCP2 and H3K27me3. In this revision, we provided additional biochemical data which support direct interaction between them as suggested by other reviewers as well. We do believe that MeCP2 functional study on other histone proteins would be extensive project in the future.

To further respond to the reviewer's concern, we stated in the manuscript:

(Line 393 - 400 of page 14, original manuscript and Line 485 of page 17, current revision) "MeCP2 has the ability to bind histone H1, H3, and H4-nucleosomal proteins as shown in Figure 1B and 1C. ~~The interaction between other~~ Other histones and their modifications could also contribute to ~~transcriptional regulation.~~ MeCP2 binding to chromatin. Indeed, a large proportion of chromatin is modified by more than one epigenetic modification, which suggests that there are complex parameters for controlling the MeCP2 binding affinity of to the whole genome and for gene expression regulation. Therefore, it is necessary to study the interaction with other epigenetic modulations in correlation with MeCP2 to achieve better understanding of MeCP2 function in the regulation of gene expression."

The study failed to provide direct evidence that MeCP2 physically binds to H3K27me3 (as depicted in Fig 5D).

Response: As stated above, we provided biochemical evidences that MeCP2, indeed, directly binds more to H3K27me3 enriched nucleosome compared to non-modified H3 nucleosome. Please see Fig. 3. This was the same concern of Reveiwer1, comment1 and Reveiwer2 comment7. We have now included additional data to support that MeCP2 directly binds to H3K27me3 enriched nucleosomes. Please see our response to Reviewer 1.

Given MeCP2's known association with DNA and chromatin, it is not surprising to observe co-localization between MeCP2 and certain histone modifications in some parts of the genome. A more important question is how MeCP2 is recruited to each locus in response to what signals. Although the study describes many correlations between MeCP2 binding, histone modification, nucleosome location, and gene expression, no novel mechanistic insights are provided.

Response: In the original manuscript, we demonstrated that genome-wide MeCP2 binding is highly correlated with H3K27me3 enriched loci. In this revision, we provide evidence that MeCP2 directly bind to H3K27me3 enriched nucleosomes by *in vitro* pull-down assay and *in vivo immunoprecipitation* assay. Taken together, we demonstrate that the mechanical interaction between specific histone PTM and MeCP2 regulates MeCP2 binding in chromatin throughout the genome.

Results from the drug treatment are not adequate because of potential side effects and indirect actions.

Response: We agree that most of experiments, if not all, using drugs always raise an issue about unnecessary side-effects. Nonetheless, GSK343 is one of the drugs which has been extensively using in related fields. It has been fully characterized as inhibitor of the histone lysine methyltransferase EZH2. In particular, GSK343 is highly selective for EZH2, which is 1000-fold more than most other methyltransferases (Verma *et al.*, 2012). In addition, GSK343 has been used up to date to control activity of EZH2 (Mohammad *et al.*, 2017, Nature Medicine; Zhou *et al.*, 2019, Nature Communications).

However, if GSK343 indirectly regulates DNA methylation levels as well as H3K27me3, changes in MeCP2 binding to chromatin may be partly the result of DNA methylation changes. To avoid this confusion, we compared the level of DNA methylation between DMSO and GSK343 treated SH-SY5Y cells in following experiment. First, we evaluate the overall change of methylated cytosine in the context of CpG dinucleotides located at LINE-1. Bisulfite – PCR pyrosequencing for LINE-1 methylation (Yang AS *et al.*, 2004) show no significant change in overall LINE-1 methylation level (Fig. 3C and Supplementary Fig. 7A and 7B). Second, we assess methylation changes at the single base resolution level of the targeted loci. Consist with the bisulfite – PCR pyrosequencing analysis, GSK343 effect on DNA methylation is very limited (Fig. 3D and Supplementary Fig. 7D). Based on these observations and other reports, we believe that GSK343 does not cause significant changes in DNA methylation relevant to the present study.

Reviewers' Comments:

Reviewer #1:

Remarks to the Author:

"MeCP2 regulates gene expression through recognition of H3K27me3" by Lee et al.

In this manuscript, Lee et al describe experimental studies and bioinformatic analysis of genome-wide co-localization between MeCP2, H3K27me3, and histone H1 to argue that MeCP2 directly binds histone H3K27me3 and that co-occurrence of MeCP2 and H3K27me3 at the TSS and gene body could regulate gene expression. This is a revised version; the original manuscript had been criticized by all reviewers as not providing sufficient mechanistic insights for direct interaction between MeCP2 and H3K27me3. In the revision, the authors sought to address this issue by pulling down either unmodified or modified nucleosomes and analyzing the bound MeCP2 protein by western blotting. However, the data resulting from this important experiment are still not convincing, especially since there were no defined motifs in MeCP2 for H3K27me3 binding nor any mutations altering the MeCP2-histone interactions. MeCP2/H3K27me3 nucleosome binding is shown as one western blot (Fig. 3A) without any quantitation and statistics. In order to make a more significant advance in understanding of gene regulation by MeCP2, the authors should provide some more profound evaluations of specific interaction of MeCP2 with nucleosomes containing H3K27me3 histone – e.g. by measuring MeCP2 affinities with nucleosomes by gel-mobility assays (see Nikitina et al. J. Biol. Chem. 2007 Sep 21;282(38):28237-45) or by titration with H3K27me3 and WT peptides to compete with nucleosome binding.

My other significant concern is that the evidence for MeCP2 and H3K27me3 cooperation in gene regulation are based on H3K27me3 enrichment at the MeCP2-regulated genes in KO mice and thus are also indirect. A more functionally-relevant and direct study should address some specific genes regulated by MeCP2 and H3K27me3. For example, some genes that requires MeCP2 for repression are expected to be de-repressed and loss its MeCP2 enrichment by knockdown or inhibiting of Ezh2 histone methyltransferase.

additional minor points:

- page 5, lane 92 "Both H3 and H4 were found to be co-precipitated with MeCP2 (Fig. 1C)." The H4 western signal on Fig. 1C is very weak. H4 presence binding needs to be confirmed e.g. by mass-spec.

- page 9, lane 215: " we performed in vivo immunoprecipitation assays" – immunoprecipitation is not an in vivo assay as it analyzes an extract from lysed cells.

Reviewer #2:

Remarks to the Author:

The authors have adequately addressed the reviewer's comments. The revised manuscript is suitable for publication.

Reviewer #3:

Remarks to the Author:

1) In reference to my original request for the authors to map out the domain of MeCP2 protein specifically responsible for its physical binding to H3K27me3, it is a critical piece of evidence to

support arguably the only novelty of this paper. Since MeCP2 has been demonstrated to physically bind to a long list of proteins, the identification of the specific binding domain has true mechanistic implications. If MeCP2 uses the same domain to bind to two different proteins, we at least know that those two partner proteins will never bind MeCP2 at the same time and location. It is unsatisfactory for the authors to simply brush that comment aside. Given that many of the MeCP2 truncation/deletion constructs already exist in the field, this is not a difficult experiment at all.

2) The "in vivo" experiments in Figure 3E were carried out using a cell line, but not in neurons or neural tissues. One narrow strip of IP-Western from OE tissue shown in Supplementary Figure 5C is of very poor quality. Those limited data severely limit the physiological significance of the conclusion (i.e. does this physical interaction really happen in neurons?). Endogenous physical interaction in neurons or neural tissues is required here.

3) One serious technical problem with all the experiments (Figure 3A and 3E in particular) purported to demonstrate physical interaction between MeCP2 and H3K27me3 is that there is no quantification (across sufficient number of biological replications) and statistical analysis of how much the interaction is reduced by GSK343 treatment.

4) Another serious technical problem is clearly demonstrated by the MeCP2 Western results in the two input lanes in Supplementary Figure 5B. It is difficult to believe any experiment performed using this anti-MeCP2 antibody given the number of nonspecific bands on that Western blot.

5) One more internal inconsistency between the data can be observed in Figure 3G, which shows the change of MeCP2 ChIP-seq signals at three types of genomic loci after GSK343 treatment. First, the changes of MeCP2 signals in the "moderate" and "severe" groups are not that big (if one compares this with the change of IP-Western results, one can see that the change in IP-Western is much bigger. By the way, this is actually another strong reason for the quantification of IP-Western results.). The statistically significant increase of MeCP2 ChIP-seq signal averaged across all H3K27me3 "unchanged" loci makes no sense at all, if MeCP2 binding to H3K27me3 is the driving factor for its binding to chromatin. Second, it makes no sense to average the MeCP2 ChIP-seq signals across all the loci, because in each category, there may be large number of loci where the change of MeCP2 ChIP-seq signal could be contrary to the loss of H3K27me3). The authors should provide stratification of MeCP2 ChIP-seq data in each group into decrease, no change, and increase for each putative H3K27me3 loci.

6) When RNA knockdown experiments are so commonly carried out nowadays, it is not acceptable for the authors to not even attempt to knockdown H3K27 methyltransferase as an independent method to validate results from the GSK343 treatment.

7) Finally, I didn't realize the data (Figure 4 and Supplementary Figure 8) showing "MeCP2 binds to H3K27me3 enriched loci independent of DNA methylation" were done in cancer cell lines (HCT116). Important conclusions like this one shouldn't be supported by only data from cancer cell line.

Reviewer #1 (Remarks to the Author):

“MeCP2 regulates gene expression through recognition of H3K27me3” by Lee et al. In this manuscript, Lee et al describe experimental studies and bioinformatic analysis of genome-wide co-localization between MeCP2, H3K27me3, and histone H1 to argue that MeCP2 directly binds histone H3K27me3 and that co-occurrence of MeCP2 and H3K27me3 at the TSS and gene body could regulate gene expression.

This is a revised version; the original manuscript had been criticized by all reviewers as not providing sufficient mechanistic insights for direct interaction between MeCP2 and H3K27me3. In the revision, the authors sought to address this issue by pulling down either unmodified or modified nucleosomes and analyzing the bound MeCP2 protein by western blotting. However, the data resulting from this important experiment are still not convincing, especially since there were no defined motifs in MeCP2 for H3K27me3 binding nor any mutations altering the MeCP2-histone interactions.

Response: We appreciate Reviewer 1’s thoughtful comments. To which domain of MeCP2 plays a central role for interacting with H3K27me3, we generated a series of deletion mutants of MeCP2. MeCP2 full length and mutants were purified and binding affinity to unmodified and H3K27me3 modified nucleosomes were carefully compared. Through this study, we observed that MBD is a key domain that preferentially interacts with H3K27me3 modified nucleosomes. The experiments were repeated and outcome is consistent. This data is shown in Fig. 3B to 3D and addressed it in the results section. Please, see line 194 - 206 of page 8 in current revision.

MeCP2/H3K27me3 nucleosome binding is shown as one western blot (Fig. 3A) without any quantitation and statistics. In order to make a more significant advance in understanding of gene regulation by MeCP2, the authors should provide some more profound evaluations of specific interaction of MeCP2 with nucleosomes containing H3K27me3 histone – e.g. by measuring MeCP2 affinities with nucleosomes by gel-mobility assays (see Nikitina et al. J. Biol. Chem. 2007 Sep 21;282(38):28237-45) or by titration with H3K27me3 and WT peptides to compete with nucleosome binding.

Response: To further address MeCP2 binding affinity to H3K27me3, we performed binding competition assay suggested by reviewer 1. Consistent with our results that MBD domain has capacity to bind to H3K27me3, MeCP2 pull-down efficiency of H3K27me3-nucleosome was clearly dampened by adding H3K27me3 peptide but not by unmodified peptide (Fig. 3E). Quantified binding efficiency is represented as graph in Fig. 3G. This result adds additional support for MeCP2/H3K27me3 interaction which is shown both by genomic and biochemical approaches. We appreciate the reviewer’s concrete suggestion. Please, see line 207 - 221 of page 8 in current revision. EMSA analysis was also attempted but was technically impossible in our lab, possibly due to high pH of MeCP2 (pH(I), 10.4), histone H3 (pH(I), 12.2), and H3K27me3 (pH(I), 11.7).

My other significant concern is that the evidence for MeCP2 and H3K27me3 cooperation in gene regulation are based on H3K27me3 enrichment at the MeCP2-regulated genes in KO mice and thus are also indirect.

A more functionally-relevant and direct study should address some specific genes regulated by MeCP2 and H3K27me3. For example, some genes that requires MeCP2 for repression are expected to be de-repressed and loss its MeCP2 enrichment by knockdown or inhibiting of Ezh2 histone methyltransferase.

Response: We discussed in the manuscript that MeCP2 interaction with methylated DNA and modified histone is complex and may involve competitive interactions (see discussion, line 488 - 496 of page 17 in current revision). The discovery of MeCP2 interaction with modified histone (H3K27me3) provide the first glimpse of the already complicated MeCP2 – chromatin interactions. This manuscript is focused on establishing firm ground of MeCP2/H3K27me3 binding. Total of 43 NGS libraries, both *in vivo* and *in vitro* evidence support our findings. We understand the reviewer's wish to better understand the precise role of MeCP2/H3K27me3 interaction in transcription. As we discover and reported that manipulation of H3K27me3 will tilt the balance of MeCP2 interaction with existing binding partners, like methylated DNA loci (see discussion, line 512 - 523 of page 18 in current revision). Due to the complexity and balance of MeCP2 interactions with methylated DNA or other proteins along the chromatin, careful, thorough and innovative multiple approaches will be required to address gene expression regulation by cooperative interactions between MeCP2 and H3K27m3. Hence, we think it is beyond the scope of this study. We hope that Reviewer 1 could understand the scope and the novelty of our current manuscript.

additional minor points:

- page 5, lane 92 “Both H3 and H4 were found to be co-precipitated with MeCP2 (Fig. 1C).” The H4 western signal on Fig. 1C is very weak. H4 presence binding needs to be confirmed e.g. by mass-spec.

(Reviewer1 only Fig. 1).

Response: Figure 1C serves to validate that MeCP2 specifically interacts with nucleosome components. Western blotting confirms MeCP2 interaction with nucleosome unit including the presence of H4. This result has been repeated multiple times, though the H4 band might appear weaker compared to other bands like H3, possibly due to the molecular size and the position on the gel (in our hands), the positive signal is evidence with the raw data shown here

- page 9, lane 215: “ we performed in vivo immunoprecipitation assays” – immunoprecipitation is not an in vivo assay as it analyzes an extract from lysed cells.

Response: We edited the sentence in the text as shown below (line 241 - 242 of page 9 in current revision). “Given these findings, we performed *in vivo* immunoprecipitation assays to determine direct MeCP2 and H3K27me3 interaction.”

Reviewer #3 (Remarks to the Author):

1) In reference to my original request for the authors to map out the domain of MeCP2 protein specifically responsible for its physical binding to H3K27me3, it is a critical piece of evidence to support arguably the only novelty of this paper. Since MeCP2 has been demonstrated to physically bind to a long list of proteins, the identification of the specific binding domain has true mechanistic implications. If MeCP2 uses the same domain to bind to two different proteins, we at least know that those two partner proteins will never bind MeCP2 at the same time and location. It is unsatisfactory for the authors to simply brush that comment aside. Given that many of the MeCP2 truncation/deletion constructs already exist in the field, this is not a difficult experiment at all.

Response: We appreciate reviewer 3's comments. To satisfy the reviewer, we performed structure-function analysis to determine domains involved in MeCP2/H3K27me3 interaction. This was the same concern of reviewer1, comment1. Please see our response to Reviewer 1.

2) The "in vivo" experiments in Figure 3E were carried out using a cell line, but not in neurons or neural tissues. One narrow strip of IP-Western from OE tissue shown in Supplementary Figure 5C is of very poor quality. Those limited data severely limit the physiological significance of the conclusion (i.e. does this physical interaction really happen in neurons?). Endogenous physical interaction in neurons or neural tissues is required here.

Response: SH-SY5Y is a human neuroblastoma cell and commonly used as *in vitro* models of neuronal function. In fact, in more than 1100 research papers, the SY5Y cell line was used or referred to in reference to the MeCP2 study (Yasui *et al.*, 2007, PNAS, number of citations: 374). In this manuscript, we analyzed genome-wide MeCP2 enrichment and H3K27me3 modification using ChIP-Seq in both olfactory epithelium tissue (OE) and SH-SY5Y cells. The strong correlation between MeCP2 and H3K27me3 ($r = 0.74$) from SH-SY5Y is similar to that from olfactory tissue ($r = 0.74$). Given the similarity of co-enrichment of the protein between SH-SY5Y and neural tissue, we believe that SH-SY5Y analysis provided sufficient insight. In addition, the reviewer mentions that the IP-Western from OE tissue, shown in Supplemental Figure 5F (current revision), is "very poor quality", but we believe it demonstrated definitive outcome of the Co-IP analysis.

3) One serious technical problem with all the experiments (Figure 3A and 3E in particular) purported to demonstrate physical interaction between MeCP2 and H3K27me3 is that there is no quantification (across sufficient number of biological replications) and statistical analysis of how much the interaction is reduced by GSK343 treatment.

Response: Data shown in Figure 3A and 3E (Fig. 3A and 3F in current revision) are representative of repeated assays and qualitative evaluation of MeCP2/H3K27me3. MeCP2/H3K27me3 interaction is further supported by different experimental approaches. Please, see Fig. 3D and 3E.

4) Another serious technical problem is clearly demonstrated by the MeCP2 Western results in the two input lanes in Supplementary Figure 5B. It is difficult to believe any experiment performed using this anti-MeCP2 antibody given the number of nonspecific bands on that Western blot.

Response: It is unexpected that extra bands of MeCP2 observed in the two input lanes which is possibly due to a stoichiometric status on antigen and antibody interaction in Western analysis that frequently happen. However, only single band is detected in Co-IP pull down assay and the size of band is corresponded to one of band in the two input lanes. Therefore, these extra bands have no impact on the interpretation of these experimental results. Specificity of the MeCP2 antibody is fully characterized and used over our 3 different project (Lee et al., 2014; Rube et al., 2016; and supplementary Fig. 1 in current manuscript) and published study (Gatta et al., 2019; Yoo et al., 2017; Zhubi et al., 2017; Mayer et al., 2015; Hauke et al., 2008). These bands were also detected by different antibodies (Cell signaling, MeCP2 (D4F3) rabbit mAb, #3456, number of citations: 48) on the same membrane.

5) One more internal inconsistency between the data can be observed in Figure 3G, which shows the change of MeCP2 ChIP-seq signals at three types of genomic loci after GSK343 treatment. First, the changes of MeCP2 signals in the “moderate” and “severe” groups are not that big (if one compares this with the change of IP-Western results, one can see that the change in IP-Western is much bigger. By the way, this is actually another strong reason for the quantification of IP-Western results.). The statistically significant increase of MeCP2 ChIP-seq signal averaged across all H3K27me3 “unchanged” loci makes no sense at all, if MeCP2 binding to H3K27me3 is the driving factor for its binding to chromatin.

Response: There has been a misunderstanding. To investigate whether MeCP2 binding decreases when H3K27me3 decreases, we classified loci, according to the degree of H3K27me3 level changes by GSK343 treatment. We visualized the H3K27me3 differences between DMSO and GSK343 treated sample, by transforming the data onto M (Y-axis: H3K27me3 difference between DMSO and GSK343 treated sample) and A (X-axis: mean average of DMSO and GSK343 treated sample). As the mean of H3K27me3 increases, the difference of H3K27me3 also increases. The results suggest that GSK343 treatment cause a decrease in H3K27me3 where H3K27me3 are enriched. On the same loci, we compare MeCP2 binding difference among the groups. **In the loci of the “unchanged” group**, MeCP2 significantly increase, but small, in the GSK343-treated samples compared to DMSO (Wilcoxon signed-rank test, $p < *0.0001$, median of MeCP2 enrichment differences = 0.00; Fig. 3J). **In the “moderate” group**, MeCP2 binding is significantly reduced in GSK343 treated cells compared to that in the control DMSO (Wilcoxon signed-rank test, $p < *0.0001$, median of MeCP2 enrichment differences = -0.47; Fig. 3J). **In the “severe” group**, the degree of MeCP2 reduction is at a greater degree compared to that of the “moderate” group (Wilcoxon signed-rank test, DMSO vs GSK343 of severe, $p = 7.75 \times 10^{-51}$, median of MeCP2 enrichment differences = -1.5; Fig. 3J). MeCP2 binding differences are statistically evaluated and show very low p-values. **These results strongly suggest that MeCP2 binding is reduced where H3K27me3 enrichment is reduced.** We also validate the difference in selected loci using ChIP-qPCR (Supplementary Fig. 6H to J). **More importantly, it is not possible to directly compare reads per million (RPM) values, arbitrary unit, with the signal strength of WB analysis.**

Regarding significant increase of MeCP2 ChIP-seq signal in H3K27me3 “unchanged” loci, this may reflect that MeCP2 binding to chromatin is a delicate balance and dependent upon the availability of multiple binding partners, such as methylated DNA. But a detailed analysis of this is beyond the scope of this study.

Second, it makes no sense to average the MeCP2 ChIP-seq signals across all the loci, because in each category, there may be large number of loci where the change of MeCP2 ChIP-seq signal could be contrary to the loss of H3K27me3. The authors should provide stratification of MeCP2 ChIP-seq data in each group into decrease, no change, and increase for each putative H3K27me3 loci.

Response: The approach taken here is a well-established bioinformatics approach. To demonstrate MeCP2 binding difference between DMSO and GSK343 treatment, we **did not** average the change of MeCP2 ChIP-seq signal in each category for statistical analysis. We used **Wilcoxon signed rank test** which is a non-parametric statistical hypothesis test, used **to compare matched samples**, such as “before” and “after”. Here, we think that, in Fig. 3J (this revision), current statistical methods are appropriate because we intend to compare MeCP2 enrichment for each locus between DMSO and GSK343 treated samples. As is, the statistical method is clearly stated in result and extended experimental procedures. In addition, since GSK343 treatment effectively inhibits H3K27me3 modification, the loci in which H3K27me3 was increased by GSK343 treatment was not observed (Please, see Fig. 3I of current revision).

6) When RNA knockdown experiments are so commonly carried out nowadays, it is not acceptable for the authors to not even attempt to knockdown H3K27 methyltransferase as an independent method to validate results from the GSK343 treatment.

Response: We respect the reviewer’s thoughts on taking knockdown approach. To optimize the experimental condition for H3K27me3 reduction, we compared the efficiency of H3K27me3 reduction induced by siRNA and pharmaceutical treatment. As the reviewer noticed in Reviewer3 only Fig. 2, while siRNA to EZH2 often does not allow achieving enough reduction of H3K27me3 levels, GSK343 treatment consistently induce reduction of H3K27me3 levels. Delayed reduction of H3K27me3 in EZH2 knockdown experiments has also been reported in previous study (Fig. 1E, Ito et al., Cell Rep. 2018 Mar 27;22(13):3480-3492)

Most importantly, we observed significant cell loss in the repeated experiments for 72 hours of incubation time after EZH2-siRNA transfection. After inducing a decrease in H3K27me3, we considered a series of experiments, including immunoprecipitation, ChIP-Seq and ChIP-qPCR validation under uniform reduction condition of H3K27me3, and the knockdown technique was excluded due to the technical problem. We believe the pharmacological approach to decrease H3K27me3 level is the most consistent and valid approach. In addition, we provided evidence that GSK343 treatment effectively reduces H3K27me3 without changing DNA methylation through biochemistry assay and genome-wide analysis. Please, see Supplementary Fig. 6 and Supplementary Fig. 7.

7) Finally, I didn't realize the data (Figure 4 and Supplementary Figure 8) showing "MeCP2 binds to H3K27me3 enriched loci independent of DNA methylation" were done in cancer cell lines (HCT116). Important conclusions like this one shouldn't be supported by only data from cancer cell line.

Response: It is technically difficult to maintain hypomethylated cells with DNMTs knocked out because DNA methylation is absolutely required for somatic cell survival. Please, see the following studies.

- Li E. et al., "Targeted mutation of the DNA methyltransferase gene results in embryonic lethality," *Cell*. 1992 Jun 12;69(6):915-26.
- Fan G. et al., "DNA hypomethylation perturbs the function and survival of CNS neurons in postnatal animals," *J Neurosci*. 2001 Feb 1;21(3):788-97.
- Okano M. et al., "DNA methyltransferases Dnmt3a and Dnmt3b are essential for de novo methylation and mammalian development," *Cell*. 1999 Oct 29;99(3):247-57.
- Liao J. et al., "Targeted disruption of DNMT1, DNMT3A and DNMT3B in human embryonic stem cells," *Nat Genet*. 2015 May;47(5):469-78

MeCP2 to bind not only methylated CpG but also unmethylated CpG in SH-SY5Y (Dag et al., 2007). The same results were confirmed in the re-analysis of five previously reported genome-wide studies, including neurons (Skene et al., 2010), ESC (Baubec et al., 2013), hypothalamus (Chen et al., 2015), forebrain (Gabel et al., 2015) and olfactory epithelium (Rube et al., 2016). On the other hand, MeCP2 binding to chromatin was highly correlated with H3K27me3 in OE tissues ($r = 0.74$), SH-SY5Y ($r = 0.74$), and HCT116 ($r=0.79$). Including our findings in HCT116, both methylation-independent MeCP2 binding and correlation between MeCP2 and H3K27m3 appears to be common regardless of cell type. Therefore, using HCT116 to study MeCP2 is unlikely to compromise the significance of our study. The manuscript already contains in vivo data that MeCP2 is located at the H3K27me3 enriched loci independent of DNA methylation. Please, see Fig.5E to H and line 361-369 in page 13 (current revision).

Reviewers' Comments:

Reviewer #1:

Remarks to the Author:

Overall, this work shows an even higher complexity in MeCP2-chromatin interactions than it was thought before and provides a valuable resource for genomic mapping of MeCP2. In the second revision the authors have provided additional experimental evidences suggesting that MeCP2 interacts with H3K27me3 through its MBD domain and performed a binding competition assay showing an interference of the H3K27me3 peptide but not of unmodified peptide with MeCP2 binding. They have also provided satisfactory explanations to other reviewer's comments. Therefore, I believe that the revised manuscript is appropriate for publication.

Reviewer #3:

Remarks to the Author:

Below is a list of my comment on revision 1, and my respective comments (bracketed by *) on the authors response.

1) In reference to my original request for the authors to map out the domain of MeCP2 protein specifically responsible for its physical binding to H3K27me3, it is a critical piece of evidence to support arguably the only novelty of this paper. Since MeCP2 has been demonstrated to physically bind to a long list of proteins, the identification of the specific binding domain has true mechanistic implications. If MeCP2 uses the same domain to bind to two different proteins, we at least know that those two partner proteins will never bind MeCP2 at the same time and location. It is unsatisfactory for the authors to simply brush that comment aside. Given that many of the MeCP2 truncation/deletion constructs already exist in the field, this is not a difficult experiment at all.

The authors carried out experiments to map the domain on MeCP2 that is physically binding to histone H3. There are four issues with those new experiments. First, quantification in Fig 3F and 3G were not normalized against GST-MeCP2 signals in Fig 3E. Second, MBD is listed in Fig 3B, but not shown in Fig 3C. Third, although MBD wasn't shown in Fig 3C, it was fused with GST and used in Fig 3D and 3E. Fourth, the label in Fig 3F seems to be wrong, because it is identical to that in Fig 3G. By the way, why is the GST alone band so strong in those GST-MBD lanes in Fig 3E? Does that fusion protein get cleaved easily? How does that cleavage affect binding from experiment to experiment?

2) The "in vivo" experiments in Figure 3E were carried out using a cell line, but not in neurons or neural tissues. One narrow strip of IP-Western from OE tissue shown in Supplementary Figure 5C is of very poor quality. Those limited data severely limit the physiological significance of the conclusion (i.e. does this physical interaction really happen in neurons?). Endogenous physical interaction in neurons or neural tissues is required here.

The authors' response is not acceptable. Endogenous physical interaction in neurons or neural tissue is required here, because, among many caveats, physiological expression level of MeCP2 is critical for protein-protein interactions. If the authors want to claim relevance to Rett syndrome, SH-SY5Y cells

are not sufficient. Citing $r=0.74$ in ChIP-seq data between OE and SH-SY5Y is not sufficient, because what if the genomic loci that are truly relevant to Rett syndrome are those where OE and SH-SY5Y differ?

In addition, my comment about the "very poor quality" of the narrow strip of IP-Western stands on its own, because that narrow strip showed non-specific H3K27me3 band in the middle lane (IgG) where there isn't supposed to be any signal. If one looks across the entire paper, nowhere else is non-specific H3K27me3 band present. Since the authors adamantly claims the quality of their data in this figure is good, I demand to see the full blot of that figure.

Finally, my comment about physiological relevance applies to ChIP-seq experiments as well.

3) One serious technical problem with all the experiments (Figure 3A and 3E in particular) purported to demonstrate physical interaction between MeCP2 and H3K27me3 is that there is no quantification (across sufficient number of biological replications) and statistical analysis of how much the interaction is reduced by GSK343 treatment.

The authors didn't experimentally answer my question about how much (quantitatively) the interaction between H3 is reduced by GSK343 treatment. This question is highly relevant to the interpretation of MeCP2 ChIP-seq change in response to GSK 343. For instance, if as shown in the middle blot anti-Histone H3 doesn't pull down MeCP2 anymore, how would one expect the MeCP2 ChIP-seq change to be in response to GSK343 treatment? Therefore, Fig3H needs to be quantified (both reciprocal IP-Western blots)

4) Another serious technical problem is clearly demonstrated by the MeCP2 Western results in the two input lanes in Supplementary Figure 5B. It is difficult to believe any experiment performed using this anti-MeCP2 antibody given the number of nonspecific bands on that Western blot.

The authors response is scientifically invalid. As shown on the Western blot in question, there are many non-specific bands. It is very reasonable to question the results generated using this anti-MeCP antibody. Given the authors claim "only single band is detected in Co-IP pull down assay", I now request the authors to provide immunoprecipitation data from either brain tissue or neurons using this antibody that demonstrate a single band (best would be without Western blotting with another MeCP2 antibody). By the way, the number of prior publications using a non-specific antibody doesn't validate the specificity of the antibody.

5) One more internal inconsistency between the data can be observed in Figure 3G, which shows the change of MeCP2 ChIP-seq signals at three types of genomic loci after GSK343 treatment. First, the changes of MeCP2 signals in the "moderate" and "severe" groups are not that big (if one compares this with the change of IP-Western results, one can see that the change in IP-Western is much bigger. By the way, this is actually another strong reason for the quantification of IP-Western results.). The

statistically significant increase of MeCP2 ChIP-seq signal averaged across all H3K27me3 “unchanged” loci makes no sense at all, if MeCP2 binding to H3K27me3 is the driving factor for its binding to chromatin. Second, it makes no sense to average the MeCP2 ChIP-seq signals across all the loci, because in each category, there may be large number of loci where the change of MeCP2 ChIP-seq signal could be contrary to the loss of H3K27me3). The authors should provide stratification of MeCP2 ChIP-seq data in each group into decrease, no change, and increase for each putative H3K27me3 loci.

Based on the authors’ response, the authors didn’t understand my comment. First, I commented that “The statistically significant increase of MeCP2 ChIP-seq signal in the H3K27me3 unchanged loci makes no sense at all, if MeCP2 binding to H3K27me3 is the driving factor for its binding to chromatin”. The authors need to respond to this comment directly.

Second, for the “moderate” and “severe” groups, I have the same question that applies to both groups. Let me elaborate using the “moderate” group in Fig 3J in the second revision manuscript. Are the green dots in the DMSO group and the GSK343 group represent the identical genomic loci, meaning for each locus Xi in DMSO, there is an identical point in GSK343? If the answer is yes, does MeCP2 binding/enrichment change between the DMSO condition and the GSK343 condition identical for each locus Xi? In another word, did MeCP2 enrichment decrease for each locus Xi in response to GSK343? If not, by stratification, I asked the authors to show how many (%) loci showed increased/unchanged/decreased MeCP2 enrichment, and who are they.

Finally, my physiological relevance comment should apply to the relevant ChIP-seq experiments as well, because the assumption that whatever biology happens in tumor cell lines will happen in neurons and brain tissues is not scientifically valid for Rett syndrome until proven otherwise.

6) When RNA knockdown experiments are so commonly carried out nowadays, it is not acceptable for the authors to not even attempt to knockdown H3K27 methyltransferase as an independent method to validate results from the GSK343 treatment.

If the H3K27 methyltransferase is specific for H3K27, why would its knockdown be more toxic than the GSK343 treatment? Doesn’t that mean either the GSK343 is not sufficiently blocking H3K27 methylation or it is acting via other targets? That being said, I understand there is little the authors can do if the cells died with siRNAi knockdown. But at least these caveats should be discussed.

7) Finally, I didn’t realize the data (Figure 4 and Supplementary Figure 8) showing “MeCP2 binds to H3K27me3 enriched loci independent of DNA methylation” were done in cancer cell lines (HCT116). Important conclusions like this one shouldn’t be supported by only data from cancer cell line.

Again, this is a big conclusion. Experimental data from neurons or brain tissues are required for relevance to Rett syndrome. Citing Figs 5E and 5F is not sufficient, because those are data from a different tumor cell line.

Response to Reviewer #1:

Overall, this work shows an even higher complexity in MeCP2-chromatin interactions than it was thought before and provides a valuable resource for genomic mapping of MeCP2. In the second revision the authors have provided additional experimental evidences suggesting that MeCP2 interacts with H3K27me3 through its MBD domain and performed a binding competition assay showing an interference of the H3K27me3 peptide but not of unmodified peptide with MeCP2 binding. **They have also provided satisfactory explanations to other reviewer's comments.** Therefore, I believe that the revised manuscript is appropriate for publication.

Response: We appreciate reviewer #1 positive feedback. We think our manuscript has been greatly improved through Reviewer 1's valuable comments.

Response to Reviewer #3:

After two rounds of revision, this manuscript has been greatly improved by the two new findings. First, MeCP2 directly interacts with H3K27me3 enriched nucleosome genome-wide. Second, MeCP2 preferentially recognizes H3K27me3 via methyl CpG binding domain (MBD). Although mutations in *MECP2* are responsible for human disease, Rett Syndrome, our study focuses on the understanding of the molecular mechanisms of MeCP2-chromatin interaction and its impact on global pattern of gene expression using model systems, including mice and cell lines. Fundamental understandings of molecular interactions are successfully acquired through the utilization of model systems and in using reductionist approaches. The novel MeCP2-H3K27me3 interaction discovered here will provide insight for designing future studies to understand MeCP2 functions in neuronal context. We believe the major misunderstanding from Reviewer 3 is precisely the reluctance in accepting the above-mentioned widely accepted investigative strategies.

Following points are the statements made by Reviewer 3 emphasizing relevance to Rett syndrome; however, we **never** state about Rett syndrome relevance with our study anywhere in the manuscript. Rather this study is mainly about genome wide regulation of gene expression by MeCP2 through nucleosome context specifically through H3K27me3.

- “If the authors want to claim relevance to Rett syndrome, SH-SY5Y cells are not sufficient. Citing $r=0.74$ in ChIP-seq data between OE and SH-SY5Y is not sufficient, because what if the genomic loci that are truly relevant to Rett syndrome are those where OE and SH-SY5Y differ?” (from reviewer3 comment #2)
- “Finally, my physiological relevance comment should apply to the relevant ChIP-seq experiments as well, because the assumption that whatever biology happens in tumor cell lines will happen in neurons and brain tissues is not scientifically valid for Rett syndrome until proven otherwise.” (from reviewer3 comment #5)
- “Experimental data from neurons or brain tissues are required for relevance to Rett syndrome.” (from reviewer3 comment #7)

We have **no intention** of claiming that this article is directly addressing the etiology of Rett syndrome. Through fundamental biochemical and genomic studies, our findings provide significant information on MeCP2's capacity to interact with modified histone in addition to methylated DNA.

Comment #1) The authors carried out experiments to map the domain on MeCP2 that is physically binding to histone H3. There are four issues with those new experiments. **First**, quantification in Fig 3F and 3G were not normalized against GST-MeCP2 signals in Fig 3E. **Second**, MBD is listed in Fig 3B, but not shown in Fig 3C. **Third**, although MBD wasn't shown in Fig 3C, it was fused with GST and used in Fig 3D and 3E. **Fourth**, the label in Fig 3F seems to be wrong, because it is identical to that in Fig 3G. By the way, why is the GST alone band so strong in those GST-MBD lanes in Fig 3E? Does that fusion protein get cleaved easily? How does that cleavage affect binding from experiment to experiment?

Response to comment#1: The structure-binding experiment was requested by both reviewers 1 and 3. We honored the reviewers' request and performed extensive truncations of MeCP2 protein and assayed for the binding ability to H3K27me3. Reviewer 1 was satisfied with the experimental design and the results. However, reviewer3 is less satisfied, so we give the following additional responses:

First, the purpose of quantification in Fig. 3F and 3G is the measurement of pulled down H3K27me3 signal changes after binding competition assay. Here, we normalize both H3K27me3 (#) and H3K27me3 peptide (##) signals with H3K27me3 intensity pulled down from non-peptide competition (lane 4 in Fig. 3E). However, reviewer 3 request us to normalize the pulled down H3K27me3 to GST-MeCP2 MBD. We, therefore, carried out the quantitative analysis (reviewer 3 only Fig.1 shown below). As seen in the figure, the trend of pulled down H3K27me3 signal change is not altered by normalization of GST-MeCP2 MBD. Because we believe that our initial normalization method is reasonable and agreed by reviewer 1 as well, and the results were not changed by the method proposed by the reviewer3, **we will keep the original figures as it is.**

Reviewer 3 only Fig. 1

Regarding Reviewer 3's second, and the third comment, the purpose of the serial experiments is to identify the domain of MeCP2 protein specifically responsible for its physical binding to H3K27me3. In order to narrow down the binding domain, initially, we mapped out the domain of MeCP2 in Fig. 3C and find that MBD containing domain is potentially critical for the physical interaction. In Fig. 3D, we further narrowed down the binding domain to MBD and validated the interaction in Fig. 3E. For clarity, all domain information in Figure 3B is required and useful for readers of this manuscript. As we stated it clearly in the result, so **we want to keep it as it is.**

Fourth, the labels in both Fig 3F and 3G are **correct.** We performed a binding competition assay and demonstrated while the H3K27me3 peptide interfered with the physical binding of MBD to H3K27me3 containing nucleosomes (FIG. 3G), but the unmodified peptide did not (FIG. 3F). **Please, see X-axis label in both Fig. 3F and 3G.**

Under our experimental condition, GST is partially cleaved. This phenomenon is well known (Fig. 6A in Dipankar Bhandari *et al.*, Genes Dev. 2014 Apr 15; 28(8): 888–901). The results of normalizing GST-MeCP2 MBD (Reviewer 3 only, Figure 1) are nearly identical to the original analysis (Figures 3F and 3G), suggesting that partial GST cleavage does not affect our results and interpretation.

Comment #2) The authors' response is not acceptable. Endogenous physical interaction in neurons or neural tissue is required here, because, among many caveats, physiological expression level of MeCP2 is critical for protein-protein interactions. If the authors want **to claim relevance to Rett syndrome**, SH-SY5Y cells are not sufficient. Citing $r=0.74$ in ChIP-seq data between OE and SH-SY5Y is not sufficient, because what if the **genomic loci that are truly relevant to Rett syndrome** are those where OE and SH-SY5Y differ?

In addition, my comment about the “very poor quality” of the narrow strip of IP-Western stands on its own, because that narrow strip showed non-specific H3K27me3 band in the middle lane (IgG) where there isn't supposed to be any signal. If one looks across the entire paper, nowhere else is non-specific H3K27me3 band present. Since the authors adamantly claims the quality of their data in this figure is good, I demand to see the full blot of that figure. Finally, my comment about physiological relevance applies to ChIP-seq experiments as well.

Response to comment #2: In regarding the “demand” for full-length blot for the IP study, we now include the full-length blot in Supplementary Fig. 5F. In this experiment, a rabbit antibody against MeCP2 is used for IP and subsequently, a rabbit antibody against H3K27me3 is used to detect associated level by western blotting. As a result, in addition to H3K27me3, we are also detecting rabbit IgG in the IP mix on the blot. Other than that, no other bands were observed on the blot as shown by the full-length blot.

Comment #3) The authors didn't experimentally answer my question about how much (quantitatively) the interaction between H3 is reduced by GSK343 treatment. This question is highly relevant to the interpretation of MeCP2 ChIP-seq change in response to GSK 343. For instance, if as shown in the middle blot anti-Histone H3 doesn't pull down MeCP2 anymore, how would one expect the MeCP2 ChIP-seq change to be in response to GSK343 treatment? Therefore, Fig3H needs to be quantified (both reciprocal IP-Western blots)

Response to comment #3: The actual density measurements of western blots are often not linear to protein abundance (Butler *et al.*, 2019). The nonlinear densitometry data were observed in both chemiluminescence and infrared fluorescence. For this reason, we prefer not to estimate signal differences in WB using density measurements. In particular, the quantification is inevitably varied according to the film exposure time on the WB membrane of the Co-IP experiment even in the same experiment. Indeed, few quantitative and statistical analyses in immunoprecipitation assays are represented in other studies.

To support our findings made from the WB analysis, we analyzed genome-wide H3K27me3 and MeCP2 enrichment using ChIP-seq (Fig. 3I and 3J) and confirmed it in the selected loci (Supplementary Fig. 6H to 6J). Because the ChIP-seq analysis provides information on the difference in both MeCP2 and H3K27me3 enrichment according to the loci by GSK343 treatment, ChIP-seq study provides better accuracy and reproducibility in measuring changes in MeCP2 binding as H3K27me3 levels change. In particular, we used the spike-in control to improve the accuracy of H3K27me3 reduction measurement by GSK343 treatment (Please, see Supplementary Fig. 6D and 6F, line 252 of the page 10, and method).

Comment #4) The authors response is scientifically invalid. As shown on the Western blot in question, there are many non-specific bands. It is very reasonable to question the results generated using this anti-MeCP2 antibody. Given the authors claim “only single band is detected in Co-IP pull down assay”, I now request the authors to provide immunoprecipitation data from either brain tissue or neurons using this antibody that demonstrate a single band (best would be without Western blotting with another MeCP2 antibody). By the way, the number of prior publications using a non-specific antibody doesn't validate the specificity of the antibody.

Response to comment #4: As stated at the beginning of the response, this study focuses on the understanding of the capacity of MeCP2-H3K27me3 interactions in general. We have already demonstrated the specificity of the MeCP2 antibody, Supplementary Fig. 1A (MeCP2 band does not detected in MeCP2 KO cells which obviously demonstrates the specificity of MeCP2 antibody) of this study and our prior studies (Lee *et al.*, 2014; Rube *et al.*, 2016), in addition to many other published studies.

Comment #5) Based on the authors' response, the authors didn't understand my comment. **First**, I commented that “The statistically significant increase of MeCP2 ChIP-seq signal in the H3K27me3 unchanged loci makes no sense at all, if MeCP2 binding to H3K27me3 is the driving factor for its binding to chromatin”. The authors need to respond to this comment directly. **Second**, for the “moderate” and “severe” groups, I have the same question that applies to both groups. Let me elaborate using the “moderate” group in Fig 3J in the second revision manuscript. Are the green dots in the DMSO group and the GSK343 group represent the identical genomic loci, meaning for each locus Xi in DMSO, there is an identical point in GSK343? If the answer is **yes**, does MeCP2 binding/enrichment change between the DMSO condition and the GSK343 condition identical for each locus Xi? In another word, did MeCP2 enrichment decrease for each locus Xi in response to GSK343? If not, by stratification, I asked the authors to show how many (%) loci showed increased/unchanged/decreased MeCP2 enrichment, and who are they. **Finally**, my physiological relevance comment should apply to the relevant ChIP-seq experiments as well, because the assumption that whatever biology happens in tumor cell lines will happen in neurons and brain tissues is **not scientifically valid for Rett syndrome** until proven otherwise.

Response to comment #5: Regarding to the first comment, though we did observe statistical significance within the “unchanged group”, **the median of MeCP2 enrichment differences is 0.00** (Fig. 3J). A statistically significant increase in MeCP2 binding may reflect a subtle redistribution of MeCP2 binding when H3K27me3 is significantly reduced. Further in-depth studies will be needed to address this subtle change of binding which is beyond the scope of this study.

Regarding the second comment, the answer is “YES”. MeCP2 binding changes between DMSO and GSK343 treatment were evaluated at the same loci where the H3K27me3 differences were compared. We already clearly stated it in results and extended methods. Also, the MeCP2 enrichment difference is analyzed using the Wilcoxon signed rank test meaning that the MeCP2 enrichment differences were statistically evaluated within the same loci. In short, the point made by reviewer 3 is already stated in line 271 of the page 10.

Comment #6) If the H3K27 methyltransferase is specific for H3K27, why would its knockdown be more toxic than the GSK343 treatment? Doesn't that mean either the GSK343 is not sufficiently blocking H3K27 methylation or it is acting via other targets? That being said, I understand there is little the authors can do if the cells died with siRNAi knockdown. But at least these caveats should be discussed.

Response to comment #6: To the best of our knowledge, there are no reports to support that GSK343 regulates H3K27me3 through other pathways or affects methylation of other targets. In particular, the effects of GSK343 on DNA methylation were carefully evaluated in biochemical experiments and genome-wide analysis using bisulfite sequencing analysis, and demonstrated no significant changes in the levels of DNA methylation. We think we have addressed this appropriately in the supplement Fig 7A to 7F.

Comment #7) Again, this is a big conclusion. Experimental data from neurons or brain tissues are required for relevance to Rett syndrome. Citing Figs 5E and 5F is not sufficient, because those are data from a different tumor cell line.

Response to comment #7: Figs 5E and 5F are characterization of epigenetic modification around TSS (-1kb to 4kb) **of neuronal tissue (mouse olfactory epithelium)**. To make it clear, we stated in the figure legend, methods and the result section (Please, see line 334 of the page 12, and line 1242 of the page 44).